# Learning explanations that are hard to vary

**Giambattista Parascandolo**[1,2,*]   **Alexander Neitz**[1,*]
**Antonio Orvieto**[2]   **Luigi Gresele**[1,3]   **Bernhard Schölkopf**[1,2]
[1]MPI for Intelligent Systems, Tübingen,   [2]ETH, Zürich,   [3]MPI for Biological Cybernetics, Tübingen
*equal contribution

## Abstract

In this paper, we investigate the principle that *good explanations are hard to vary* in the context of deep learning. We show that averaging gradients across examples – akin to a logical OR ($\vee$) of patterns – can favor memorization and 'patchwork' solutions that sew together different strategies, instead of identifying invariances. To inspect this, we first formalize a notion of consistency for minima of the loss surface, which measures to what extent a minimum appears only when examples are pooled. We then propose and experimentally validate a simple alternative algorithm based on a logical AND ($\wedge$), that focuses on invariances and prevents memorization in a set of real-world tasks. Finally, using a synthetic dataset with a clear distinction between invariant and spurious mechanisms, we dissect learning signals and compare this approach to well-established regularizers.

## 1 Introduction

Consider the top of Figure 1, which shows a view from above of the loss surface obtained as we vary a two dimensional parameter vector $\theta = (\theta_1, \theta_2)$, for a fictional dataset containing two observations $x_A$ and $x_B$. Note the two global minima on the top-right and bottom-left. Depending on the initial values of $\theta$ — marked as white circles — gradient descent converges to one of the two minima. Judging solely by the value of the loss function, which is zero in both cases, the two minima look equally good.

However, looking at the loss surfaces for $x_A$ and $x_B$ separately, as shown below, a crucial difference between those two minima appears: Starting from the same initial parameter configurations and following the gradient of the loss, $\nabla_\theta \mathcal{L}(\theta, x_i)$, the probability of finding the same minimum on the top-right in either case is zero. In contrast, the minimum in the lower-left corner has a significant overlap across the two loss surfaces, so gradient descent can converge to it even if training on $x_A$ (or $x_B$) only. Note that after averaging there is no way to tell what the two loss surfaces looked like: *Are we destroying information that is potentially important?*

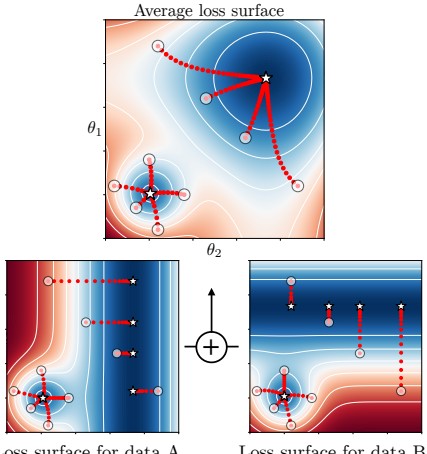

Figure 1: Loss landscapes of a two-parameter model. Averaging gradients forgoes information that can identify patterns shared across different environments.

In this paper, we argue that the answer is yes. In particular, we hypothesize that if the goal is to find *invariant mechanisms* in the data, these can be identified by finding explanations (e.g. model parameters) that are hard to vary across examples. A notion of invariance implies something that stays the same, as something else changes. We assume that data comes from different *environments*: An invariant mechanism is shared across all, generalizes out of distribution (o.o.d.), but might be hard to model; each environment also has spurious explanations that are easy to spot ('shortcuts'), but do not generalize o.o.d. From the point of view of causal modeling, such invariant mechanisms can be interpreted as conditional distributions of the targets given causal features of the inputs; invariance of such conditionals is expected if they represent *causal mechanisms*, that is — stable properties of the physical world (see e.g. Hoover (1990)). Generalizing o.o.d. means therefore that the predictor should perform equally well on data coming from different settings, as long as they share the causal mechanisms.

We formalize a notion of *consistency*, which characterizes to what extent a minimum of the loss surface appears *only* when data from different environments are pooled. Minima with low consistency are 'patchwork' solutions, which (we hypothesize) sew together different strategies and should not be expected to generalize to new environments. An intuitive description of this principle was proposed by physicist David Deutsch: *"good explanations are hard to vary"* (Deutsch, 2011).

Using the notion of consistency, we define *Invariant Learning Consistency* (ILC), a measure of the expected consistency of the solution found by a learning algorithm on a given hypothesis class. The ILC can be improved by changing the hypothesis class or the learning algorithm, and in the last part of the paper we focus on the latter. We then analyse why current practices in deep learning provide little incentive for networks to learn invariances, and show that standard training is instead set up with the explicit objective of greedily maximizing speed of learning, i.e., progress on the training loss. When learning "as fast as possible" is not the main objective, we show we can trade-off some "learning speed" for prioritizing learning the invariances. A practical instantiation of ILC leads to o.o.d. generalization on a challenging synthetic task where several established regularizers fail to generalize; moreover, following the memorization task from Zhang et al. (2017), ILC prevents convergence on CIFAR-10 with random labels, as no shared mechanism is present, and similarly when a portion of training labels is incorrect. Lastly, we set up a behavioural cloning task based on the game CoinRun (Cobbe et al., 2019b), and observe better generalization on new unseen levels.

**An example.** Take these two second-hand books of chess puzzles. We can learn the two independent shortcuts (blue arrows for the left book *OR* handwritten solutions on the right), or actually learn to play chess (the invariant mechanism). While both strategies solve other problems from the same books (i.i.d.), only the latter generalises to new chess puzzle books (o.o.d.). How to distinguish the two? We would not have learned about the red arrows had we trained on the book on the right, and vice versa with the hand-written notes.

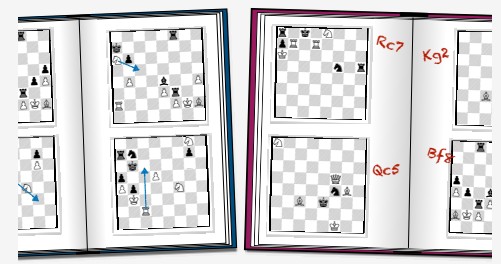

## 2 EXPLANATIONS THAT ARE HARD TO VARY

We consider datasets $\{\mathcal{D}^e\}_{e \in \mathcal{E}}$, with $|\mathcal{E}| = d$, and $\mathcal{D}^e = (x_i^e, y_i^e)$, $i_e = 1, \ldots, n^e$. Here $x_i^e \in \mathcal{X} \subseteq \mathbb{R}^m$ is the vector containing the observed inputs, and $y_i^e \in \mathcal{Y} \subseteq \mathbb{R}^p$ the targets. The superscript $e \in \mathcal{E}$ indexes some aspect of the data collection process, and can be interpreted as an environment label. Our objective is to infer a function $f : \mathcal{X} \rightarrow \mathcal{Y}$ — which we call *mechanism* — assigning a target $y_i^e$ to each input $x_i^e$; as explained in the introduction, we assume that such function is shared across all environments. For estimation purposes, $f$ may be parametrized by a neural network with continuous activations; for weights $\theta \in \Theta \subseteq \mathbb{R}^n$, we denote the neural network output at $x \in \mathcal{X}$ as $f_\theta(x)$.

**Gradient-based optimization.** To find an appropriate model $f_\theta$, standard optimizers rely on gradients from a *pooled* loss function $\mathcal{L} : \mathbb{R}^n \rightarrow \mathbb{R}$. This function measures the *average* performance of the neural network when predicting data labels, across all environments: $\mathcal{L}(\theta) := \frac{1}{|\mathcal{E}|} \sum_{e \in \mathcal{E}} \mathcal{L}_e(\theta)$, with $\mathcal{L}_e(\theta) := \frac{1}{|\mathcal{D}^e|} \sum_{(x_i^e, y_i^e) \in \mathcal{D}^e} \ell(f(x_i^e; \theta), y_i^e)$; where $\ell : \mathbb{R}^p \times \mathbb{R}^p \rightarrow [0, +\infty)$ is usually chosen to be the $L2$ loss or the cross-entropy loss. The parameter updates according to gradient descent (GD) are given by $\theta_{\text{GD}}^{k+1} = \theta_{\text{GD}}^k - \eta \nabla \mathcal{L}(\theta_{\text{GD}}^k)$, where $\eta > 0$ is the learning rate. Under some standard assumptions (Lee et al., 2016), $(\theta_{\text{GD}}^k)_{k \geq 0}$ converges to a local minimizer of $\mathcal{L}$, with probability one.

**When do we *not* learn invariances?** We start by describing what might prevent learning invariances in standard gradient-based optimization.

*(i) Training stops once the loss is low enough.* If optimization learned spurious patterns by the time it converged, invariances will not be learned anymore. This depends on the rate at which different patterns are learned. The rates at which invariant patterns emerge (and vice-versa, the spurious patterns do not) can be improved by e.g.: (a) careful architecture design, e.g. as done by hardcoding spatial equivariance in convolutional networks; (b) fine-tuning models pre-trained on large amounts of data, where strong features already emerged and can be readily selected.

*(ii) Learning signals: everything looks relevant for a dataset of size 1.* Due to the summation in the definition of the pooled loss $\mathcal{L}$, gradients for each example are computed independently. Informally, each signal is identical to the one for an equivalent dataset of size 1, where every pattern appears relevant to the task. To find invariant patterns across examples, if we compute our training signals on each of them independently, we have to rely on the way these are aggregated.[1]

*(iii) Aggregating gradients: averaging maximizes learning speed.* The default method to pool gradients is the *arithmetic mean*. GD applied to $\mathcal{L}$ is designed to minimize the pooled loss *by prioritizing descent speed*.[2] Indeed, a step of GD is equivalent to finding a tight[3] quadratic upper bound $\hat{\mathcal{L}}$ to $\mathcal{L}$, and then jumping to the minimizer of this approximation (Nocedal and Wright, 2006). While speed is often desirable, by construction GD ignores one potentially crucial piece of information: The gradient $\nabla\mathcal{L}$ is the result of averaging signals $\nabla\mathcal{L}_e$, which correspond to the patterns visible from each environment at this stage of optimization. In other words, GD with average gradients greedily maximizes for learning speed, but in some situations we would like to trade some convergence speed for invariance.

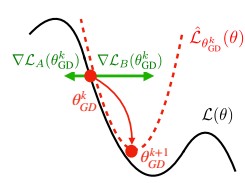

For instance, instead of performing an arithmetic mean between gradients (logical OR), we might want to look towards a logical AND, which can be characterized as a *geometric* mean. Fig. 1 shows how a sum can be seen as a logical OR: the two orthogonal gradients from data $A$ and data $B$ at $(0.5, 0.5)$ point to different directions, yet both are kept in the combined gradient.[4] In Sec. 2.3 we elaborate on this idea and on implementing a logical AND between gradients. Before presenting this discussion, we take some time to better motivate the need for invariant learning consistency and to construct a precise mathematical definition of consistency.

Figure 2: Inconsistency in gradient directions.

## 2.1 FORMAL DEFINITION OF ILC

Let $\Theta_{\mathcal{A}}^*$ be the set of convergence points of algorithm $\mathcal{A}$ when trained using all environments (pooled data): that is, $\Theta_{\mathcal{A}}^* = \{\theta^* \in \Theta \mid \exists\, \theta^0 \in \mathbb{R}^n \text{ s.t. } \mathcal{A}_\infty(\theta^0, \mathcal{E}) = \theta^*\}$. For instance, if $\mathcal{A}$ is gradient descent, the result of Lee et al. (2016) implies that $\Theta_{\mathcal{A}}^*$ is the set of local minimizers of the pooled loss $\mathcal{L}$. To each $\theta^* \in \Theta_{\mathcal{A}}^*$, we want to associate a consistency score, quantifying the concept "*good $\theta^*$ are hard to vary*". In other words, we would like the score to capture the consistency of the loss landscape around $\theta^*$ across the different environments. For example, in Fig. 1 the loss landscape near the bottom-left minimizer is consistent across environments, while the top-right minimizer is not.

Let us characterize the landscape around $\theta^*$ from the perspective of a fixed environment $e \in \mathcal{E}$. We define the set $N_{e,\theta^*}^\epsilon$ to be the largest path-connected region of space containing both $\theta^*$ and the set $\{\theta \in \Theta \text{ s.t.} |\mathcal{L}_e(\theta) - \mathcal{L}_e(\theta^*)| \leq \epsilon\}$, with $\epsilon > 0$. In other words, if $\theta \in N_{e,\theta^*}^\epsilon$ then there exist a path-connected region in parameter space including $\theta^*$ and $\theta$ where each parameter also is in $N_{e,\theta^*}^\epsilon$ and its loss on environment $e$ is comparable. From the perspective of

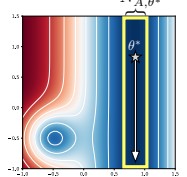
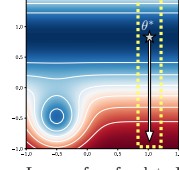

Loss surface for data A          Loss surface for data B

environment $e$, all these points are equivalent to $\theta^*$. We would like to evaluate the elements of this set with respect to a different environment $e' \neq e$. We will say that $e'$ is consistent with $e$ in $\theta^*$ if $\max_{\theta \in N_{e,\theta^*}^\epsilon} |\mathcal{L}_{e'}(\theta) - \mathcal{L}_e(\theta)|$ is small. Repeating this reasoning for all environment pairs, we arrive at the following *inconsistency score*:

$$\mathcal{I}^\epsilon(\theta^*) := \max_{(e,e')\in\mathcal{E}^2} \max_{\theta\in N_{e,\theta^*}^\epsilon} |\mathcal{L}_{e'}(\theta) - \mathcal{L}_e(\theta^*)|. \tag{1}$$

---

[1] After computing the gradients for a dataset of $n-1$ examples, if an $n$-th example appeared, we would just compute one more vector of gradients and add it to the sum. A Gaussian Process (Rasmussen, 2003) for example would require recomputing the entire solution from scratch, as all interactions are considered.

[2] The same reasoning holds for SGD in the finite-sum optimization case $\mathcal{L} = \frac{1}{m}\sum_{i=1}^m \mathcal{L}_i$, where gradients from a mini-batch are seen as unbiased estimators of gradients from the pooled loss (Bottou et al., 2018).

[3] Assume that $\mathcal{L}$ has $L$-Lipschitz gradients (i.e. curvature bounded from above by $L$). Then, at any point $\tilde{\theta}$, we can construct the upper bound $\hat{\mathcal{L}}_{\tilde{\theta}}(\theta) = \mathcal{L}(\tilde{\theta}) + \nabla\mathcal{L}(\tilde{\theta})^\top(\theta - \tilde{\theta}) + L\|\theta - \tilde{\theta}\|^2/2$.

[4] Loosely speaking, a sum is large if any of the summands is large, a product is large if all factors are large.

This consistency is our formalization of the principle "*good explanations are hard to vary*". Finally, we can write down an invariant learning consistency score for $\mathcal{A}$:

$$\text{ILC}(\mathcal{A}, p_{\theta^0}) := -\mathbb{E}_{\theta^0 \sim p(\theta^0)}\left[\mathcal{I}^\epsilon(\mathcal{A}_\infty(\theta^0, \mathcal{E}))\right]. \tag{2}$$

That is, the learning consistency of an algorithm measures the expected consistency across environments of the minimizer it converges to on the pooled data.

**Example: low consistency of a classic patchwork solution.** One-hidden-layer networks with sigmoid activations and enough neurons can approximate any function $f^* : [0, 1] \to \mathbb{R}$ (Cybenko, 1989). In appendix A.1 we show how the construction used to obtain the weights leads to a maximally inconsistent solution according to $\mathcal{I}^\epsilon(\theta^*)$, which would not be expected to generalize o.o.d.

## 2.2 ILC AS A LOGICAL AND BETWEEN LANDSCAPES

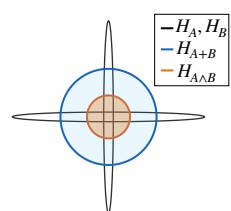

Here we draw a connection between our definition of inconsistency and the local geometric properties of the loss landscapes. For the sake of clarity, we consider two environments ($A$ and $B$) and assume $\theta^*$ to be a local minimizer (with zero loss) for both environments. Using a Taylor approximation[5], we get $\mathcal{L}(\theta) \approx \frac{1}{2}(\theta - \theta^*)^\top H_{A+B}(\theta - \theta^*)$ for $\|\theta - \theta^*\| \approx 0$, where $H_{A+B} = (H_A + H_B)/2$ is the *arithmetic mean* of the Hessians $H_A := \nabla^2 \mathcal{L}_A(\theta^*)$ and $H_B := \nabla^2 \mathcal{L}_A(\theta^*)$. $H_{A+B}$ does not capture the possibly conflicting geometries of landscape $A$ or $B$: It performs a "logical OR" on the dominant eigendirections. In contrast, the *geometric mean*, or Karcher mean, $H_{A\wedge B}$ (Ando et al., 2004) is affected by the inconsistencies between landscapes: It performs a "logical AND". In appendix A.2, we give a formal definition of $H_{A\wedge B}$, and show that for diagonal Hessians, $\mathcal{I}^\epsilon(\theta^*) \leqslant 2\epsilon(\frac{\det(H_{A+B})}{\det(H_{A\wedge B})})^2$. As

Figure 3: Plotted are contour lines $\theta^\top H^{-1}\theta = 1$ for $H_A = \text{diag}(0.05, 1)$ and $H_B = \text{diag}(1, 0.05)$. $H_{A\wedge B}$ retains the original volumes, while for $H_{A+B}$ it is $5\times$ bigger. This magnification shows inconsistency of $A$ and $B$.

for the geometric mean of positive numbers, $0 \leqslant \det(H_{A\wedge B}) \leqslant \det(H_{A+B})$; thus, inconsistency is lowest when *shapes* of $A$ and $B$ are similar – exactly as in the bottom-left minimizer of Fig. 1.

**From Hessians to gradients.** We just saw that the consistency of $\theta^*$ is linked to the geometric mean of the Hessians $\{H_e(\theta^*)\}_{e\in\mathcal{E}}$. Under the simplifying assumption that each $H_e$ is diagonal[6] and all eigenvalues $\lambda_i^e$ are positive, their geometric mean is $H^\wedge := \text{diag}((\prod_{e\in\mathcal{E}} \lambda_1^e)^{1/|\mathcal{E}|}, \dots, (\prod_{e\in\mathcal{E}} \lambda_n^e)^{1/|\mathcal{E}|})$. The curvature of the corresponding loss in the $i$-th eigendirection depends on how consistent the curvatures of each environment are in that direction. Consider now optimizing from a point $\theta^k$; gradient descent reads $\theta^{k+1} = \theta^k - \eta H^+(\theta^k - \theta^*)$, where $H^+ := \text{diag}(\frac{1}{|\mathcal{E}|}\sum_{e\in\mathcal{E}}\lambda_1^e, \dots, \frac{1}{|\mathcal{E}|}\sum_{e\in\mathcal{E}}\lambda_n^e)$. For $\eta$ small enough[7], we have $|\theta_i^{k+1} - \theta_i^*| = (1 - \eta\frac{1}{|\mathcal{E}|}\sum_{e\in\mathcal{E}}\lambda_i^e)|\theta_i^k - \theta_i^*|$. As noted, this choice maximises the speed of convergence to $\theta^*$, but does not take into account whether this minimizer is consistent. We can reduce the speed of convergence on directions where landscapes have different curvatures – which would lead to a high inconsistency – by following the gradients from the geometric mean of the landscapes, as opposed to the arithmetic mean. I.e, we substitute the full gradient $\nabla\mathcal{L}(\theta) = H^+(\theta^k - \theta^*)$ with $\nabla\mathcal{L}^\wedge(\theta) = H^\wedge(\theta^k - \theta^*)$. Also, we have that[8] $\nabla\mathcal{L}^\wedge(\theta) = (\prod_{e\in\mathcal{E}}\nabla\mathcal{L}_e(\theta))^{1/|\mathcal{E}|}$: to reduce the speed of convergence in directions with inconsistency, we can take the element-wise geometric mean of gradients from different environments (see also Fig. 11 in the appendix).

## 2.3 MASKING GRADIENTS WITH A LOGICAL AND

The element-wise geometric mean of gradients, instead of the arithmetic mean, increases consistency in the convex quadratic case. However, there are a few practical limitations:
(i) The geometric mean is only defined when all the signs are consistent. It is still to be defined how sign inconsistencies, which can occur in non-convex settings, should be dealt with.
(ii) It provides little flexibility for 'partial' agreement: Even a single zero gradient component in one environment stops optimization in that direction.

---

[5]This provides a useful simplified perspective. Indeed, this *quadratic model* is heavily used in the optimization community (see e.g. Jastrzębski et al. (2017); Zhang et al. (2019a); Mandt et al. (2017).)

[6]It was shown in (Becker et al., 1988) and recently in (Adolphs et al., 2019; Singh and Alistarh, 2020) that neural networks have a strong diagonal dominance of the Hessian matrix at the end of training.

[7]Smaller than $1/\lambda_{\max}$, $\lambda_{\max}$ is the maximum eigenvalue of Hessians from different environments,

[8]This holds if $\theta - \theta^*$ is positive, otherwise we have $\nabla\mathcal{L}^\wedge(\theta) = -\left(\prod_{e\in\mathcal{E}}|\nabla\mathcal{L}_e(\theta)|\right)^{1/|\mathcal{E}|}$.

(iii) For numerical stability, it needs to be computed in $\log$ domain (more computationally expensive).

(iv) Adaptive step-size schemes (e.g. Adam (Kingma and Ba, 2015)) rescale the signal component-wise for local curvature adaptation. The exact magnitude of the geometric mean would be ignored and most of the difference from arithmetic averaging will come from the zero-ed components.

(i) can be overcome by treating different signs as zeros, resulting in a geometric mean of 0 if there is any sign disagreement across environments for a gradient component. For (ii) we can allow for *some* disagreement (with a hyperparameter), by not masking out if there is a large percentage of environments with gradients in that direction. (iii) and (iv) can be addressed together: Since the final magnitude will be rescaled except for masked components, i.e. where the geometric mean is 0, we can use the average gradients (fast to compute) and mask out the components based on the sign agreement (computable avoiding the $\log$ domain).

**The AND-mask.** We translate the reasoning we just presented to a practical algorithm that we will refer to as the *AND-mask*. In its most simple implementation, we zero out those gradient components with respect to weights that have *inconsistent signs* across environments. Formally, the masked gradients at iteration $k$ are $m_t(\theta^k) \odot \nabla \mathcal{L}(\theta^k)$, where $m_t(\theta^k)$ vanishes for any component where there are less than $t \in \{d/2, d/2+1, \ldots, d\}$ agreeing gradient signs across environments ($d$ is the number of environments in the batch), and is equal to one otherwise. For convenience, our implementation of the AND-mask uses a *threshold* $\tau \in [0, 1]$ as hyper-parameter instead of $t$, such that $t = \frac{d}{2}(\tau + 1)$. Mathematically, for every component $[m_\tau]_j$ of $m_\tau$, $[m_\tau]_j = \mathbb{1}\left[\tau d \leqslant |\sum_e \text{sign}([\nabla \mathcal{L}_e]_j)|\right]$.

Computing the AND-mask has the same time and space complexity of standard gradient descent, i.e., linear in the number of examples that we average. Due to its simplicity and computational efficiency, this is the algorithm that we will use in the experiment section. As a first result, we show that following the AND-masked gradient leads to convergence in the directions made visible by the AND-mask. The proof is presented in appendix A.3.

*Proposition* 1. Let $\mathcal{L}$ have $L$-Lipschitz gradients and consider a learning rate $\eta \leqslant 1/L$. After $k$ iterations, AND-masked GD visits at least once a point $\theta$ where $\|m_t(\theta) \odot \nabla \mathcal{L}(\theta)\|^2 \leqslant \mathcal{O}(1/k)$.

**Behaviour in the face of randomness.** Here we put the AND mask through a theoretical test: For gradients coming from different environments that are inconsistent (or even random), how fast does the AND mask reduce the magnitude of the step taken in parameter space, compared to standard GD? *In case of inconsistency, the AND mask should quickly make the gradient steps more conservative.*

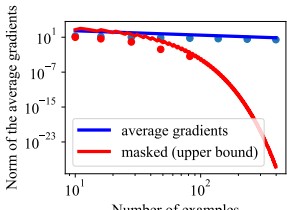

To assess this property, we consider a fixed set of $n$ parameters $\theta$ and gradients $\nabla \mathcal{L}_e$ drawn independently from a multivariate Gaussian with zero mean and unit covariance.

Figure 4: Magnitude of gradient (average or masked) on random data ($|\theta| = 3000$, $t = 0.8d$).

*Proposition* 2. Consider the setting we just outlined, with $\mathcal{L} = (1/d)\sum_{e=1}^d \mathcal{L}_e$. While $\mathbb{E}\|\nabla \mathcal{L}(\theta)\|^2 = \mathcal{O}(n/d)$, we have that $\forall t \in \{d/2+1, \ldots, d\}, \exists c \in (1, 2]$ such that $\mathbb{E}\|m_t(\theta) \odot \nabla \mathcal{L}(\theta)\|^2 \leqslant \mathcal{O}(n/c^d)$.

The proof is presented in Appendix A.4, and an illustration with numerical verification in Fig. 4 (the magnitudes of masked gradients (•) for more than 100 examples were always zero in the numerical verification). Intuitively, in the presence of purely random patterns, the AND-mask has a desirable property: it decreases the strength of these signals exponentially fast, as opposed to linearly.

## 3 EXPERIMENTS

Real-world datasets are generated by (causal) generative processes which share mechanisms (Pearl, 2009). However, mechanisms and spurious signals are often entangled, making it hard to assess what part of the learning signal is due to either. As the goal of this paper is to dissect these two components to understand how they ultimately contribute to the learning process, we create a simple synthetic dataset that allows us to control the complexity, intensity, and number of shortcuts in the data. After that, we evaluate whether spurious signals can be detected even in high-dimensional networks and datasets by testing the AND-mask on a memorization task similar to the one proposed in Zhang et al. (2017), and on a behavioral cloning task using the game CoinRun (Cobbe et al., 2019a).

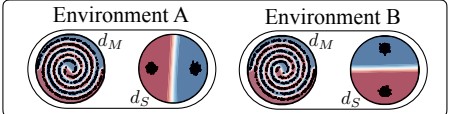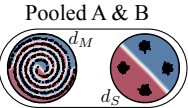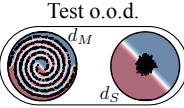

Figure 5: A 4-dimensional instantiation of the synthetic memorization dataset for visualization. Every example is a dot in both circles, and it can be classified by finding either of the "oracle" decision boundaries shown.

### 3.1  THE SYNTHETIC MEMORIZATION DATASET

We introduce a binary classification task. The input dimensionality is $d = d_M + d_S$. While $p(y|x_{d_M})$ is the same across *all* environments (i.e. the *mechanism*), $p(y|x_{d_S}, e)$ is *not the same* across all environments (the *shortcuts*). While the mechanism is shared, it needs a highly non-linear decision boundary to classify the data. The shortcuts are not shared across environments, but provide a simple way to classify the data, even when pooling all the environments together. See Figure 5 for a concrete example with $d_M$ and $d_S$ equal to 2, and two environments ($A$ and $B$). The spirals (on $d_M$) are invariant but hard to model. The shortcuts (on $d_S$) are simple blobs but different in every environment: in $A$, linearly separable through a vertical decision boundary, in $B$ with a horizontal one. If the two environments are pooled, a new diagonal decision boundary emerges on the shortcut dimensions as the most 'natural' one. While this perfectly classifies data in both environments $A$ and $B$, critically *it would have not been found by training on either partition $A$ or $B$ alone*. The out-of-distribution (o.o.d.) test data has the same mechanism but random shortcuts. Therefore, any method relying exclusively on the shortcuts will have chance-level o.o.d. performance. Details about the dataset, baselines, and training curves are reported in appendix B.

Despite the apparent simplicity of this dataset, note that it is challenging to find the invariant mechanism. In high dimensions, even with tens of pooled environments, the shortcuts allow for a *simple* classification rule under almost every classical definition of 'simple': the boundary is *linear*, it has a *large margin*, it can be expressed with *small weights*, it is *fast to learn*, robust to input noise, and has *perfect accuracy and no i.i.d. generalization gap*. Finding the complex decision boundary of the spirals, instead, is a fiddly process and arguably a much slower path towards small loss.

**Baselines.** We evaluate several domain-agnostic baselines (all multilayer perceptrons) with some of the most common regularizers used in deep learning — Dropout, L1, L2, Batch normalization. We also consider methods that explicitly make use of the environment labels, namely: *(i)* Domain Adversarial Neural Networks (DANN) (Ganin et al., 2016), a method specifically designed to address domain adaptation by obfuscating domain information with an adversarial classifier; *(ii)* Invariant Risk Minimization (IRM) (Arjovsky et al., 2019), discussed in detail in appendix B. The AND-mask is trained with the same configurations in Table 1.

**Results.** Fig. 6 shows training and test accuracy. DANN fails because it can align the representation-layer distributions from different environments using only shortcuts, such that they become indistinguishable to the domain-discriminating classifier. The AND-mask was the only method to achieve perfect test accuracy, by fitting the spirals instead of the shortcuts. In particular, the combination of the AND-mask with L1 or L2 regularization gave the most robust results overall, as they help suppress neurons that at initialization are tuned towards the shortcuts.

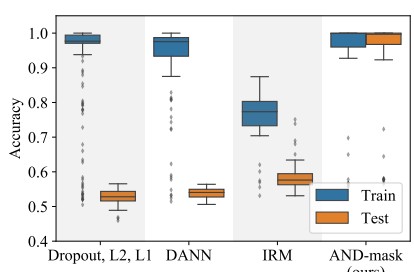

Figure 6: Results on the synthetic dataset.

**Correlations between average, memorization and generalization gradients.** Due to the synthetic nature of the dataset, we can intervene on its data-generating process in order to examine the learning signals coming from the mechanisms and from the shortcuts. We isolate the two and measure their contribution to the average gradients, as we vary the agreement threshold of the mask. More precisely, we look at the gradients computed with respect to the weights of a randomly initialized network for different sets of data: (i) The original data, with mechanisms and shortcuts. (ii) Randomly permuting the dataset over the mechanisms dimensions, thus leaving the "memorization" signal of the shortcuts. (iii) Randomly permuting over the shortcuts dimensions, isolating the "generalization" signal of the mechanisms alone. Figure 7 shows the correlation be-

tween the components of the original average gradient (i) and the shortcut gradients ((ii), dashed line), and between the original average gradients and the mechanism gradients ((iii), solid line).

While the signal from the mechanisms is present in the original average gradients (i.e. $\rho \approx 0.4$ for $\tau = 0$), its magnitude is smaller and it is 'drowned' by the memorization signal. Instead, increasing the threshold of the AND-mask (right side) suppresses memorization gradients due to the shortcuts, and for $\tau \approx 1$ most of the gradient components remaining contain signal from the mechanism. On the left side, we test the other side of our hypothesis: An XOR-mask zeroes out consistent gradients, preserves those with different signs, and results in a sharper decrease of the correlation with the mechanism gradients.

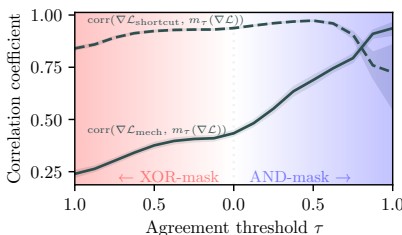

Figure 7: Gradient correlations.

## 3.2 EXPERIMENTS ON CIFAR-10

**Memorization in a vision task.** Zhang et al. (2017) showed that neural networks trained with standard regularizers — like L2 and Dropout — can still memorize large training datasets with *shuffled* labels, i.e. reaching $\approx$100% training accuracy. Their experiments raised significant questions about the generalization properties of neural networks and the role of regularizers in constraining the hypothesis class. Our hypothesis is that ILC — for example implemented as the AND-mask — should prevent memorization on a similar task with the shuffled labels, as gradients will tend to largely 'disagree' in the absence of a shared mechanism. However, when the labels are *not shuffled*, ILC should have a much weaker effect, as real shared mechanisms are still present in the data.

To test our hypothesis, we ran an experiment that closely resembles the one in (Zhang et al., 2017) on CIFAR-10. We trained a ResNet on CIFAR-10 with *random labels*, with and without the AND-mask. In all experiments we used batch size 80, and treated each example as its own "environment". Recall that standard gradient averaging is equivalent to an AND-mask with threshold 0. As shown in Figure 8, the ResNet with standard average gradients memorized the data, while slightly increasing the threshold for the AND-mask quickly prevented memorization (dark blue line). In contrast, training the same networks on the dataset *with the original labels* resulted in both of them converging and generalizing to the test set, confirming that the mask did not significantly affect the generalization error with a general underlying mechanism in the data.

Note that there is no standard notion of environments in CIFAR-10, which is why we treated every example as coming from its own environment. This assumption is not unreasonable, as every image in the dataset was literally collected in a different physical environment. If anything, it is the standard i.i.d. assumption that hides this variety behind a notion of a single distribution encompassing all environments. The results of this experiment further support this interpretation, and can serve as evidence that — in some cases — we might be able to identify invariances even without an explicit partition into environments, as this can be already identified at the level of individual examples.

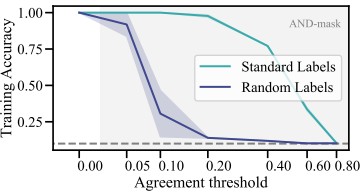

Figure 8: As the AND-mask threshold increases, memorization on CIFAR-10 with random labels is quickly hindered.

**Label noise.** Following up on this experiment, we test how the AND-mask performs in the presence of label noise, i.e. when a portion of the labels in the training set are randomly shuffled (25% here). According to our hypothesis, gradients computed on examples with random labels should disagree and get masked out by the AND-mask, while signal from correctly labeled data should contribute to update the model. As shown in Figure 9, the performance on the *incorrectly* labeled portion of the dataset is well *below* chance for the AND-mask (as it predicts correctly despite the wrong labels), while the baseline again memorizes the incorrect labels. On the test set (with untouched labels), the baseline peaks early then decreases as the model overfits, while the AND-mask slowly but steadily improves.

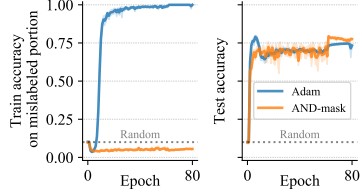

Figure 9: The AND-mask prevents overfitting to the incorrectly labeled portion of the training set (left) without hurting the test accuracy (right).

### 3.3 Behavioral Cloning on CoinRun

CoinRun (Cobbe et al., 2019b) is a game introduced to test how RL agents generalize to novel situations. The agent needs to collect coins, jumping on top of walls and boxes and avoiding enemies.[9] Each level is procedurally generated — i.e. it has a different combination of sprites, background, and layout — but the physics and goals are invariant. Cobbe et al. (2019b) showed that state-of-the-art RL algorithms fail to model these invariant mechanisms, performing poorly on new levels unless trained on thousands of them. To test our hypothesis, we set up a behavioral cloning task using CoinRun.[10] We start by pre-training a strong policy $\pi^*$ using standard PPO (Schulman et al., 2017) for 400M steps on the full distribution of levels. We then generate a dataset of pairs $(s, \pi^*(a|s))$ from the on-policy distribution. The training data consists of 1000 states from each of 64 levels, while test data comes from 2000 levels. A ResNet-18 $\hat{\pi}_\theta$ is then trained to minimize the loss $D_{\mathrm{KL}}(\pi^*||\hat{\pi}_\theta)$ on the training set. We compare the generalization performance of regular Adam to a version that uses the AND-mask. For each method we ran an automatic hyperparameter optimization study using Tree-structured Parzen Estimation (Bergstra et al., 2013) of 1024 trials.

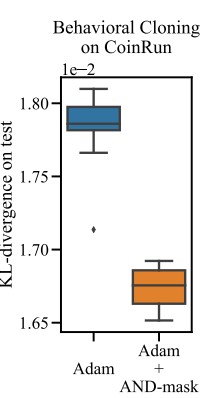

Despite the theoretical computational efficiency of computing the AND-mask as presented in Section 2.3 (i.e., linear time and memory in the size of the mini-batch, just like classic SGD), current deep learning frameworks like PyTorch (Paszke et al., 2017) have optimized routines that sum gradients across examples in a mini-batch before it is possible to efficiently compute the AND-mask. We therefore test the AND-mask in a slightly different way. In training, in each iteration we sample a batch of data from a randomly chosen level out of the 64 available (and cycle through them all once per epoch). We then apply the AND-mask 'temporally', only allowing gradients that are consistent across time (and therefore across levels). See Algorithm 1 in appendix B.6 for a detailed description of this alternative formulation of the AND-mask. The figure shows the minimum test loss for the 10 best runs, supporting the hypothesis that the AND-mask helps identify invariant mechanisms across different levels.

## 4 Related Work

*Generalization and covariate shift.* The classic formulation of statistical learning theory (Vapnik) concerns learning from independent and identically distributed samples. The case where the distribution of the covariates at test time differs from the one observed during training is termed *covariate shift* (Sugiyama et al., 2007; Quionero-Candela et al., 2009; Sugiyama and Kawanabe, 2012). Standard solutions involve re-weighting of the training examples, but require the additional assumption of overlapping supports for train and test distributions.

*Causal models and invariances.* As we mentioned in the Introduction, causality provides a strong motivation for our work, based on the notion that statistical dependencies are epiphenomena of an underlying causal model (Pearl, 2009; Peters et al., 2017). The causal description identifies stable elements – e.g. physical *mechanisms* – connecting causes and effects, which are expected to remain *invariant* under interventions or changing external conditions (Haavelmo, 1943; Schölkopf et al., 2012)). This motivates our notion of invariant mechanisms, and inspired related notions which have been proposed for robust regression (Rojas-Carulla et al., 2018; Heinze-Deml et al., 2018; Arjovsky et al., 2019; Hermann and Lampinen, 2020; Ahuja et al., 2020; Krueger et al., 2020). We discuss this in more detail in appendix C.1.

*Domain generalization.* ILC can be used in a setting of domain generalization (Muandet et al., 2013), but it is not limited to it: as demonstrated in the experiments in Section 3.2, the AND-mask can be applied even if domain labels are not available. In contrast, by treating every example as a single domain, methods relying on domain classifiers (like DANN Ganin et al. (2016) or Balaji et al. (2018)) would require as many output units as there are training examples (i.e. 50'000 for CIFAR-10).

*Gradient agreement.* Looking at gradient agreement to learn meaningful representations in neural networks has been explored in (Du et al., 2018; Eshratifar et al., 2018; Fort et al., 2019; Zhang et al., 2019b). These approaches mainly rely on a measure of cosine similarity between gradients, which

---

[9]See Figure 17 in appendix B.6 for a visualization of the game.

[10]To obtain a robust evaluation, we preferred to approach behavioral cloning instead of the full RL problem, as it is a standard supervised learning task and has substantially fewer moving parts than most deep RL algorithms.

we did not consider here for two main reasons: *(i)* It is a 'global' property of the gradients, and it would not allow us to extract precise information about different patterns in the network; *(ii)* It is unclear how to extend it beyond pairs of vectors, and for pairwise interactions its computational cost scales quadratic in the number of examples used.

## 5 CONCLUSIONS

Generalizing out of distribution is one of the most significant open challenges in machine learning, and relying on invariances across environments or examples may be key in certain contexts. In this paper we analyzed how neural networks trained by averaging gradients across examples might converge to solutions that ignore the invariances, especially if these are harder to learn than spurious patterns. We argued that if learning signals are collected *on one example at the time* — as it is the case for gradients, e.g., computed with backpropagation — the way these signals are aggregated can play a significant role in the patterns that will ultimately be expressed: Averaging gradients in particular can be too permissive, acting as a *logical OR* of a collection of distinct patterns, and lead to a 'patchwork' solution. We introduced and formalized the concept of Invariant Learning Consistency, and showed how to learn invariances even in the face of alternative explanations that — although spurious — fulfill most characteristics of a good solution. The AND-mask is but one of multiple possible ways to improve consistency, and it is unlikely to be a practical algorithm for all applications. However, we believe this should not distract from the general idea which we are trying to put forward — namely, that it is worthwhile to study learning of explanations that are *hard to vary*, with the longer term goal of advancing our understanding of learning, memorization and generalization.

## ACKNOWLEDGMENTS

We wish to thank Sebastian Gomez, Luca Biggio, Julius von Kügelgen, Paolo Penna, Ioannis Anagno, Ricards Marcinkevics, Sidak Pal Singh, Damien Teney for feedback on the manuscript, and thank Nando de Freitas for fruitful discussions in the early stage of this project. We also thank the Max Planck ETH Center for Learning Systems for supporting Giambattista Parascandolo, and the International Max Planck Research School for Intelligent Systems for supporting Alexander Neitz.

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

## A    APPENDIX TO SECTION 2

### A.1    A CLASSIC EXAMPLE OF A PATCHWORK SOLUTION

Consider a neural network with one hidden layer consisting of two neurons and sigmoidal activations:

$$f_\theta(x) = \theta_5 \sigma(\theta_1 x + \theta_2) + \theta_6 \sigma(\theta_3 x + \theta_4), \quad \sigma(z) := 1/(1 + e^{-z}). \tag{3}$$

We want to learn the continuous function $f^* : [0,1] \to [0,2]$ defined as

$$f^*(x) = \begin{cases} 0 & x \in [0, 0.4); \\ 10(x - 0.4) & x \in [0.4, 0.5); \\ 1 & x \in [0.5, 0.7); \\ 10(x - 0.7) + 1 & x \in [0.7, 0.8); \\ 2 & x \in [0.8, 1]. \end{cases}$$

To perform this task, we have access to (noiseless) data from two environments:

$$A : \{(x, f(x)) \mid x \in [0, 0.5)\}, \quad B : \{(x, f(x)) \mid x \in [0.5, 1]\}.$$

There is a simple *constructive* way, provided by the universal function approximation theorem Cybenko (1989) to fit this function[11] using $f_\theta$ up to an arbitrarily small mean squared error $\mathcal{L}_{A+B}(\theta^*)$. Leaving out the details of such a construction (Cybenko (1989) for details), the reader can check on the left panel of Figure 10 that $\theta^* = (100, -50, 100, -75, 1, 1)$ provides a good fit for *both environments* A and B — both $\mathcal{L}_A(\theta^*)$ and $\mathcal{L}_B(\theta^*)$ are small.

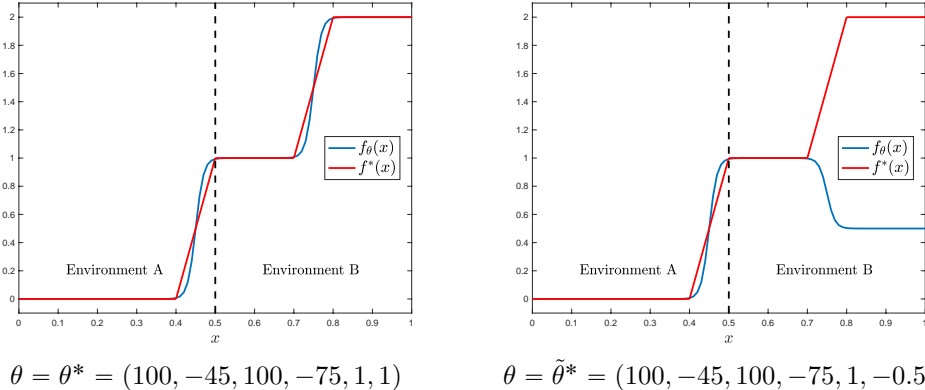

$$\theta = \theta^* = (100, -45, 100, -75, 1, 1) \qquad \theta = \tilde{\theta}^* = (100, -45, 100, -75, 1, -0.5)$$

Figure 10: Performance of the neural network in Equation 3 for two different parameters. Any reasonable modification on $\theta_6$ (say $\pm 1$) leaves the performance on environment A unchanged, while the performance on environment B quickly degrades.

However, it is easy to realize that $\theta^*$ — while being a solution which can be returned by gradient descent using the pooled data A+B — is not *consistent* (formal definition given in the main paper in Section 2). Indeed, it is possible to modify $\tilde{\theta}^*$ such that the loss in environment A remains almost unchanged, while the loss in environment B gets larger. In particular, on the right panel of Figure 10, we show that $\tilde{\theta}^* = (100, -50, 100, -75, 1, -0.5)$ is such that $\mathcal{L}_A(\theta^*) \leqslant \mathcal{L}_A(\tilde{\theta}^*) + \epsilon$ (with $\epsilon$ very small) but $\mathcal{L}_B(\theta^*) \ll \mathcal{L}_B(\tilde{\theta}^*)$. According to our definition in Equation 1 (see main paper), we have $\mathcal{I}^\epsilon(\theta^*) \leqslant |\mathcal{L}_B(\theta^*) - \mathcal{L}_B(\tilde{\theta}^*)|$ — that is a large number (low consistency).

*Remark* 1 (Connection to out of distribution generalization). The main point of this analysis was to show an example of where our measure of *consistency* behaves according to expectations: A typical implementation of the universal approximation theorem — which one would *not* expect to generalize out of distribution, due to its *'patchwork'* behavior — leads indeed to a very low consistency score.

---

[11]For a graphical description, the reader can check `http://neuralnetworksanddeeplearning.com/chap4.html`

**Geometric mean of matrices.**   Given an $n$-tuple of $d \times d$ positive definite matrices $(A_j)_{j=1}^n$, the geometric (Karcher) mean Ando et al. (2004) is the unique positive definite solution $X$ to the equation $\sum_{i=1}^m \log(A_i^{-1} X) = 0$, where $\log$ is the matrix logarithm. This matrix average has many desirable properties, which make it relevant to signal processing and medical imaging. The Karcher mean can also be written as $\arg\min_{X \in \mathcal{S}^{++}(d)} f(X) = \frac{1}{2m} \sum_{i=1}^m d(A_i, X)^2$, where $d$ is the Riemannian distance in the manifold of SPD matrices $\mathcal{S}^{++}(d)$.

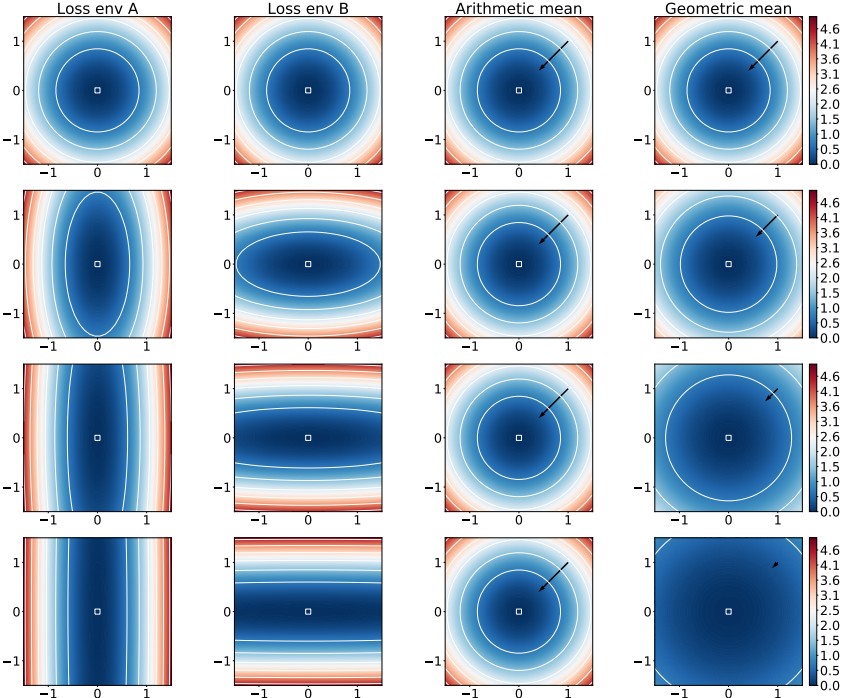

Figure 11: While the arithmetic mean of the two loss surfaces on the left is identical in all three cases (third column), the geometric mean has weaker and weaker gradients (black arrow) the more inconsistent the two loss surfaces become.

**Link between consistency and geometric means.**   Here we show how the consistency score introduced in Equation 1 can be linked (in a simplified setting) to a comparison between the arithmetic and geometric means of the Hessians approximating the landscapes of two separate environments $A$ and $B$.

At the local minimizer $\theta^* = 0$, we assume that $\mathcal{L}_A = \mathcal{L}_B = 0$ and consider the local quadratic approximations $\mathcal{L}_A(\theta) = \frac{1}{2}\theta^\top H_A \theta$ and $\mathcal{L}_B(\theta) = \frac{1}{2}\theta^\top H_B \theta$. Here, we make the additional simplifying assumption that $H_A$ and $H_B$ are diagonal (or, more broadly, co-diagonalizable): $H_A = \text{diag}(\lambda_1^A, \cdots, \lambda_n^A)$, $H_B = \text{diag}(\lambda_1^B, \cdots, \lambda_n^B)$, with $\lambda_i^A \geqslant 0$ and $\lambda_i^B \geqslant 0$ for all $i = 1, \ldots, n$. The *arithmetic* and *geometric* means (noted as $H_{A+B}$ and $H_{A \wedge B}$) of these matrices are defined in this simplified setting as follows:

$$H_{A+B} = \text{diag}\left(\frac{1}{2}(\lambda_1^A + \lambda_1^B), \cdots, \frac{1}{2}(\lambda_n^A + \lambda_n^B)\right), \quad H_{A \wedge B} = \text{diag}\left(\sqrt{\lambda_1^A \lambda_1^B}, \cdots, \sqrt{\lambda_n^A \lambda_n^B}\right).$$

As motivated in the main paper and in Figure 12, one can link the consistency of two landscapes to a comparison between the geometric and arithmetic means of the corresponding Hessians.

*Proposition* 3. In the setting we just described, the consistency score in Equation 1 can be estimated as follows:

$$\mathcal{I}^\epsilon(\theta^*) \leqslant 2\epsilon \left(\frac{\det(H_{A+B})}{\det(H_{A \wedge B})}\right)^2.$$

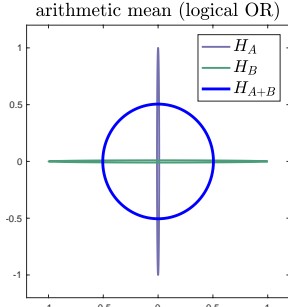
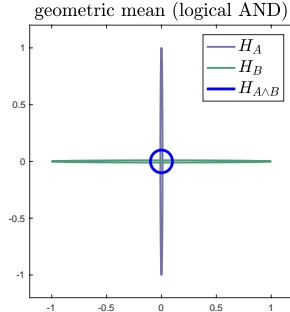

Figure 12: Plotted are contour lines $\theta^\top H^{-1}\theta = 1$ for $H_A = \mathrm{diag}(0.01, 1)$ and $H_B = \mathrm{diag}(1, 0.01)$. It is convenient to provide this visualization because it is linked to the matrix determinant: $\mathrm{Vol}(\{\theta^\top H^{-1}\theta = 1\}) = \pi\sqrt{\det(H)}$. The geometric average retains the volume of the original ellipses, while the volume of $H_{A+B}$ is 25 times bigger. This magnification indicates that landscape $A$ is not consistent with landscape $B$.

Before showing the proof, we note that the proposition gives a *lower bound* on the consistency. That is, it provides a *pessimistic* estimate. Yet, as we motivated, this estimate has a nice geometric interpretation. However, as we outline in a remark after the proof, this estimate is tight in two important limit cases.

*Proof.* In this setting, Equation 1 gives

$$\mathcal{I}^\epsilon(\theta^*) := \max\left\{\max_{\mathcal{L}_A(\theta)\leqslant\epsilon}\mathcal{L}_B(\theta), \max_{\mathcal{L}_B(\theta)\leqslant\epsilon}\mathcal{L}_A(\theta)\right\}.$$

Recall that

$$\mathcal{L}_A(\theta) = \frac{1}{2}\theta^\top H_A\theta = \frac{1}{2}\sum_i \lambda_i^A\theta_i^2.$$

Hence, this is a simple quadratic program with quadratic constraints, and

$$\max_{\mathcal{L}_A(\theta)\leqslant\epsilon}\mathcal{L}_B(\theta) = \max_{\frac{1}{2}\sum_i \lambda_i^A\theta_i^2\leqslant\epsilon}\frac{1}{2}\sum_i \lambda_i^B\theta_i^2.$$

Further, we can change variables and introduce $\tilde{\theta}_i = \theta_i\sqrt{\lambda_i^A/2}$. The problem gets even simpler:

$$\max_{\mathcal{L}_A(\theta)\leqslant\epsilon}\mathcal{L}_B(\theta) = \max_{\|\tilde{\theta}\|^2\leqslant\epsilon}\sum_i \frac{\lambda_i^B}{\lambda_i^A}\tilde{\theta}_i^2 = \epsilon\cdot\max_i\frac{\lambda_i^B}{\lambda_i^A}.$$

All in all, we get

$$\mathcal{I}^\epsilon(\theta^*) = \epsilon\max\left\{\max_i\frac{\lambda_i^B}{\lambda_i^A}, \max_i\frac{\lambda_i^A}{\lambda_i^B}\right\}$$

$$= \epsilon\cdot\max_i\max\left\{\frac{\lambda_i^B}{\lambda_i^A}, \frac{\lambda_i^A}{\lambda_i^B}\right\}$$

$$\leqslant \epsilon\cdot\max_i\left(\frac{\lambda_i^B}{\lambda_i^A} + \frac{\lambda_i^A}{\lambda_i^B}\right)$$

$$= \epsilon\cdot\max_i\left\{\frac{(\lambda_i^B)^2 + (\lambda_i^A)^2}{\lambda_i^B\lambda_i^A}\right\}$$

$$\leqslant \epsilon\cdot\max_i\left\{\frac{(\lambda_i^B + \lambda_i^A)^2}{\lambda_i^B\lambda_i^A}\right\}.$$

This means

$$\sqrt{\mathcal{I}^\epsilon(\theta^*)} \leqslant \epsilon\max_i\frac{\lambda_i^B + \lambda_i^A}{\sqrt{\lambda_i^B\lambda_i^A}} = 2\epsilon\max_i\frac{(\lambda_i^B + \lambda_i^A)/2}{\sqrt{\lambda_i^B\lambda_i^A}} \leqslant 2\epsilon\frac{\prod_i(\lambda_i^B + \lambda_i^A)/2}{\prod_i\sqrt{\lambda_i^B\lambda_i^A}} = 2\epsilon\frac{\det(H_{A+B})}{\det(H_{A\wedge B})},$$

where the first inequality comes from the monotonicity of the square root function, and the second inequality comes from the fact that (i) the geometric mean is always smaller or equal than the arithmetic mean and (ii) for any sequence of numbers $\alpha_i > 1$, $\max_i\alpha_i \leqslant \prod_i\alpha_i$. $\qquad\square$

*Remark* 2 (Sanity check). There are two important cases where we can test the bound above. First, if $H_A = H_B$, then $\mathcal{I}^\epsilon(\theta^*) = \epsilon$, and the bound returns $\mathcal{I}^\epsilon(\theta^*) \leqslant 2\epsilon$, since the geometric and arithmetic mean are the same. Next, say $\lambda_i^A = 0$ but $\lambda_i^B > 0$; then, both the bound and the inconsistency score are $\infty$ (highest possible inconsistency).

### A.3 PROOF OF PROPOSITION 1

In this appendix section we consider the AND-masked GD algorithm, introduced at the end of Section 2. We recall that the masked gradients at iteration $k$ are $m_t(\theta^k) \odot \nabla\mathcal{L}(\theta^k)$, where $m_t(\theta^k)$ vanishes for any component where there are less than $t \in \{d/2 + 1, \ldots, d\}$ agreeing gradient signs across environments, and is equal to one otherwise. In a full-batch setting, the algorithm is

$$\theta^{k+1} = \theta^k - \eta\, m_t(\theta^k) \odot \nabla\mathcal{L}(\theta^k), \qquad\qquad \text{(AND-masked GD)}$$

where $\eta > 0$ is the learning rate.

*Proposition* 1. Let $\mathcal{L}$ have $L$-Lipschitz gradients and consider a learning rate $\eta \leqslant 1/L$. After $k$ iterations, AND-masked GD visits at least once a point $\theta$ where $\|m_t(\theta) \odot \nabla\mathcal{L}(\theta)\|^2 \leqslant \mathcal{O}(1/k)$.

*Proof.* Thanks to the component-wise $L$-smoothness and using a Taylor expansion around $\theta^i$ we have

$$\mathcal{L}(\theta^{i+1}) \leqslant \mathcal{L}(\theta^i) - \eta\langle\nabla\mathcal{L}(\theta^i), m_t(\theta^i) \odot \nabla\mathcal{L}(\theta^i)\rangle + \frac{L\eta^2}{2}\|m_t(\theta^i) \odot \nabla\mathcal{L}(\theta^i)\|^2$$

$$= \mathcal{L}(\theta^i) - \left(\eta - \frac{L\eta^2}{2}\right)\|m_t(\theta^i) \odot \nabla\mathcal{L}(\theta^i)\|^2.$$

If we seek $\eta - L\eta^2/2 \geqslant \eta/2$, then $\eta \leqslant \frac{1}{L}$, as we assumed in the proposition statement. Therefore, $\mathcal{L}(\theta^{i+1}) \leqslant \mathcal{L}(\theta^i) - (\eta/2)\|m_t(\theta^i) \odot \nabla\mathcal{L}(\theta^i)\|^2$, for all $i \geqslant 0$. Summing over $i$ from 0 to a desired iteration $k$, we get

$$\sum_{i=0}^{k-1}(\eta/2)\|m_t(\theta^i) \odot \nabla\mathcal{L}(\theta^i)\|^2 \leqslant \mathcal{L}(\theta^0) - \mathcal{L}(\theta^k) \leqslant \mathcal{L}(\theta^0).$$

Therefore,

$$\min_{i=0,\ldots,k}\|m_t(\theta^i) \odot \nabla\mathcal{L}(\theta^i)\|^2 \leqslant \frac{1}{k}\sum_{i=0}^{k-1}(\eta/2)\|m_t(\theta^i) \odot \nabla\mathcal{L}(\theta^i)\|^2 \leqslant \frac{2\mathcal{L}(\theta^0)}{\eta k}.$$

Hence, there exist an iteration $i^* \in \{0, \ldots, k\}$ such that $\|m_t(\theta^{i^*}) \odot \nabla\mathcal{L}(\theta^{i^*})\|^2 \leqslant \mathcal{O}(1/k)$. $\qquad\square$

### A.4 PROOF OF PROPOSITION 2

Here we fix parameters $\theta \in \mathbb{R}^n$ and assume gradients $\nabla\mathcal{L}_e(\theta) \in \mathbb{R}^n$ coming from environments $e \in \mathcal{E}$ are drawn independently from a multivariate Gaussian with zero mean and $\sigma^2 I$ covariance. We want to show that, in this random setting, the AND-mask introduced in Section 2.3 decreases the magnitude of the gradient step.

*Proposition* 2. Consider the setting we just outlined, with $\mathcal{L} = (1/d)\sum_{e=1}^d \mathcal{L}_e$. While $\mathbb{E}\|\nabla\mathcal{L}(\theta)\|^2 = \mathcal{O}(n/d)$, we have that $\forall t \in \{d/2 + 1, \ldots, d\}, \exists c \in (1, 2]$ such that $\mathbb{E}\|m_t(\theta) \odot \nabla\mathcal{L}(\theta)\|^2 \leqslant \mathcal{O}(n/c^d)$.

*Proof.* Let us drop the argument $\theta$ for ease of notation. First, let us consider $\nabla\mathcal{L}$ (no gradient AND-mask):

$$\mathbb{E}\left\|\frac{1}{d}\sum_{i=1}^d \nabla\mathcal{L}_{e_i}\right\|^2 = \frac{1}{d^2}\sum_{i=1}^d \mathbb{E}\|\nabla\mathcal{L}_{e_i}\|^2 = \frac{n\sigma^2}{d},$$

where in the first equality we used the fact that the $\nabla\mathcal{L}_{e_i}$ are uncorrelated and in the second the fact that $\mathbb{E}[\|\nabla\mathcal{L}_{e_i}\|^2]$ is the trace of the covariance of $\nabla\mathcal{L}_{e_i}$.

Next, assume we apply the element-wise AND-mask $m_t$ to the gradients, which puts to zero the components (dimensions) where there are less than $t \in \{d/2, \ldots, d\}$ equal signs. Since Gaussians are symmetric around zero, the probability of having *exactly* $u$ positive $j$-th gradient component among $d$ environments is $Pr(p_j = u) = \left(\frac{1}{2}\right)^d \binom{d}{u}$. Hence, the probability to keep the $j$-th gradient direction (considering also negative consistency) is

$$
\begin{aligned}
\Pr[[m_t]_j = 1] &= \sum_{u=t}^{d} \Pr(p_j = u) + \sum_{u=0}^{d-t} \Pr(p_j = u) \\
&= \left(\frac{1}{2}\right)^d \sum_{k=t}^{d} \binom{d}{k} + \left(\frac{1}{2}\right)^d \sum_{k=0}^{d-t} \binom{d}{k} \\
&= 2 \left(\frac{1}{2}\right)^d \sum_{k=t}^{d} \binom{d}{k}.
\end{aligned}
\tag{4}
$$

We would now like to compute $\mathbb{E} \left\| m_t \odot \left(\frac{1}{d} \sum_{i=1}^{d} \nabla \mathcal{L}_{e_i}\right) \right\|^2$. The difficulty lies in the fact that the event $m_t = 1$ makes gradients conditionally *dependent*. Indeed, conditioning on both $m_t = 1$ and $[\nabla \mathcal{L}_e]_j > 0$ changes the distribution of $[\nabla \mathcal{L}_{e'}]_j$: this gradient entry is going to be more likely to be positive or negative, depending on the value of $[\nabla \mathcal{L}_e]_j$ and on the details of the gradient mask. To solve the issue, we our strategy is to reduce the discussion (without loss in generality and with no additional assumption) to the case where gradient entries have all the same sign and hence conditional independence is restored.

We consider the following writing for the quantity we are interested in:

$$
\begin{aligned}
\mathbb{E} \left\| m_t \odot \left(\frac{1}{d} \sum_{i=1}^{d} \nabla \mathcal{L}_{e_i}\right) \right\|^2 &= \sum_{j=1}^{n} \mathbb{E} \left[ [m_t]_j \left(\frac{1}{d} \sum_{i=1}^{d} [\nabla \mathcal{L}_{e_i}]_j\right)^2 \right] \\
&= \sum_{j=1}^{n} \sum_{\hat{p}_j=0}^{d} \mathbb{E} \left[ [m_t]_j \left(\frac{1}{d} \sum_{i=1}^{d} [\nabla \mathcal{L}_{e_i}]_j\right)^2 \bigg| p_j = \hat{p}_j \right] \Pr[p_j = \hat{p}_j] \\
&= \sum_{j=1}^{n} \sum_{\hat{p}_j=0}^{(d-t)} \sum_{\hat{p}_j=t}^{d} \mathbb{E} \left[ \left(\frac{1}{d} \sum_{i=1}^{d} [\nabla \mathcal{L}_{e_i}]_j\right)^2 \bigg| p_j = \hat{p}_j \right] \Pr[p_j = \hat{p}_j] \\
&= 2 \sum_{j=1}^{n} \sum_{\hat{p}_j=t}^{d} \mathbb{E} \left[ \left(\frac{1}{d} \sum_{i=1}^{d} [\nabla \mathcal{L}_{e_i}]_j\right)^2 \bigg| p_j = \hat{p}_j \right] \left(\frac{1}{2}\right)^d \binom{d}{\hat{p}_j},
\end{aligned}
$$

where we used the definition of 2-norm, the law of total expectation, and the symmetry of the problem with respect to positive and negative numbers. Finally, since the gradient components within the same environment are conditionally independent, for any $j \in \{1, \ldots, n\}$ we can write

$$
\mathbb{E} \left\| m_t \odot \left(\frac{1}{d} \sum_{i=1}^{d} \nabla \mathcal{L}_{e_i}\right) \right\|^2 = 2n \sum_{\hat{p}_j=t}^{d} \mathbb{E} \left[ \left(\frac{1}{d} \sum_{i=1}^{d} [\nabla \mathcal{L}_{e_i}]_j\right)^2 \bigg| p_j = \hat{p}_j \right] \left(\frac{1}{2}\right)^d \binom{d}{\hat{p}_j}.
$$

Finally, we note that the following bound holds:

$$
\mathbb{E} \left[ \left(\frac{1}{d} \sum_{i=1}^{d} [\nabla \mathcal{L}_{e_i}]_j\right)^2 \bigg| p_j = \hat{p}_j \leqslant d \right] \leqslant \mathbb{E} \left[ \left(\frac{1}{d} \sum_{i=1}^{d} [\nabla \mathcal{L}_{e_i}]_j\right)^2 \bigg| p_j = d \right].
$$

Indeed, if *all* environments lead to positive (or, symmetrically, negative) and *non-interacting* gradients in the $j$-th direction, the average will be the biggest in norm. Moreover — crucially — conditioned on the event $p_j = d$, gradients coming from different environments are distributed as a positive half-normal distributions. Moreover, they are *conditionally independent*; this because, since they are

all positive, the value of a gradient in one environment cannot influence the value of the gradient in another one. We remark that conditional independence on the right-hand side is therefore *not an assumption*, but is intrinsic to the upper bound.

Putting it all together, we have

$$\mathbb{E}\left\|m_t \odot \left(\frac{1}{d}\sum_{i=1}^{d}\nabla\mathcal{L}_{e_i}\right)\right\|^2 \leqslant 2n\sum_{\hat{p}_j=t}^{d}\mathbb{E}\left[\left(\frac{1}{d}\sum_{i=1}^{d}[\nabla\mathcal{L}_{e_i}]_j\right)^2\bigg|p_j=d\right]\left(\frac{1}{2}\right)^d\binom{d}{\hat{p}_j}$$

$$\leqslant 2n\sum_{\hat{p}_j=t}^{d}\sigma^2\left(\frac{1}{2}\right)^d\binom{d}{\hat{p}_j}$$

$$\leqslant \sigma^2 n(d-t)\binom{d}{t}\left(\frac{1}{2}\right)^{d-1},$$

where in the second line we bounded the squared average of a sum of half normal distributions: let $\{X_i\}_{i=1}^d$ be a family of uncorrelated positive half-normal distributions derived from a Gaussians with mean zero and variance $\sigma^2$, we have[12] that $\mathbb{E}[X_i]=\sigma\sqrt{2/\pi}$ and $\mathbb{E}[X_i^2]=\sigma^2$. Also, $\mathbb{E}[X_iX_j]=\mathbb{E}[X_i]\mathbb{E}[X_j]\leqslant\sigma^2$. Therefore,

$$\mathbb{E}\left[\left(\frac{1}{d}\sum_{i=1}^{d}X_i\right)^2\right]=\frac{1}{d^2}\sum_{i,j=1}^{d}\mathbb{E}[X_iX_j]\leqslant\sigma^2.$$

Finally, if we set $r=t/d\in(0.5,1]$, we have[13]

$$\binom{d}{t}\sim\left(\frac{1}{r^r(1-r)^{1-r}}\right)^d$$

as $d\to\infty$ (discarding all polynomial terms). Hence $\binom{d}{t}$ is of the form $q^d$, with $1\leqslant q<2$. So, the quantity $\sigma^2 n(d-t)\binom{d}{t}\left(\frac{1}{2}\right)^{d-1}$ will be exponentially decreasing at a rate $\mathcal{O}(n/(2-q)^d)$. Notably, if $t=d/2$, then we lose the exponential rate and get back to $\mathcal{O}(n/d)$. □

---

[12]https://en.wikipedia.org/wiki/Half-normal_distribution
[13]Theorem 1 in Burić, Tomislav, and Neven Elezović. "Asymptotic expansions of the binomial coefficients." Journal of applied mathematics and computing 46.1-2 (2014): 135-145.

# B  APPENDIX TO SECTION 3

We used Pytorch Paszke et al. (2017) to implement all experiments in this paper. Our codebase is publicly available at `https://github.com/gibipara92/learning-explanations-hard-to-vary`.

## B.1  SECTION 3.1

Table 1: Hyperparameter ranges for synthetic data experiments. The regularizers L1 and L2 are never combined; instead, one weight regularization type out of L1, L2 and none is selected and we sample from the respective range afterwards.

| Hyperparameter | Ranges |
|---|---|
| No. hidden units | $\{256, 512\}$ |
| No. hidden layers | $\{3, 5\}$ |
| Batch-size | $\{64, 128, 256\}$ |
| Optimizer | $\{\text{Adam}_{\beta_1=0.9, \beta_2=0.999}, \text{SGD} + \text{momentum}_{0.9}\}$ |
| Learning rate | $\{\text{1e-3, 1e-2, 1e-1}\}$ |
| Batch-normalization | $\{\text{Yes, No}\}$ |
| Dropout | $\{0.0, 0.5\}$ |
| L2 regularization | $\{\text{1e-5, 1e-4, 1e-3}\}$ |
| L1 regularization | $\{\text{1e-6, 1e-5, 1e-4}\}$ |

## B.2  DATASET

Here we report more technical details about the synthetic dataset described in Section 3. Each example is constructed as follows: we first choose the label randomly to be either $+1$ or $-1$, with equal probability. The example is a vector with $d_S + d_M$ entries, consisting of the *shortcut* and the *mechanism*. In our experiments, $d_M = 2$ and $d_S = 32$.

The Gaussian *shortcuts* are obtained by first sampling one random vector $\mathbf{x}_s \in \mathbb{R}^{d_S}$ per environment. Its components $x_{s,i}$ are sampled independently from a Normal distribution: $x_{s,i} \sim \mathcal{N}(0, 0.1)$. We use $\mathbf{x}_s$ for class 1, and $-\mathbf{x}_s$ for class -1. In the test set, all shortcut components are sampled i.i.d. from the same Normal distribution. Effectively, each example of the test set belongs to a different domain. The *mechanism* is implemented as the two interconnected spirals shown in Figure 13 by sampling the radius $r \sim \text{Unif}(0.08, 1.0)$ and then computing the angle as $\alpha = 2\pi nr$ where $n$ is the number of revolutions of the spiral. We add uniform noise in the range $[-0.02, 0.02]$ to the radii afterwards.

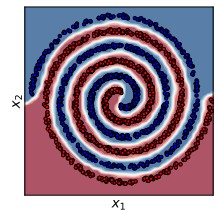

Figure 13: The spirals used as the *mechanism* in the synthetic memorization dataset.

The training dataset consists of 1280 examples per environment and we use $D = 32$ environments unless otherwise mentioned. The training datasets consists of 2000 examples.

## B.3  EXPERIMENT

We train all networks for $\lceil 3000/D \rceil$ epochs, dropping the learning rate by a factor 10 halfway through, and again at three-quarters of training. For computational reason, we stop each trial before completion if the training accuracy exceeds 97% and the test accuracy is below 60%. All networks are MLPs with LeakyReLU activation functions and a cross-entropy loss on the output. We run a hyperparameter search over the ranges shown in Table 1. For IRM and the AND-mask, we select the best-performing run and re-run it 50 times with different random seeds. For DANN and the standard baselines nothing produced results significantly better than chance.

### B.3.1  STANDARD REGULARIZERS AND AND-MASK

The networks with the L1, L2, Dropout and Batch-normalization regularizers, have hyperparameters that were randomly selected from Table 1. For the AND-mask we used the very same ranges. The

regularizers L1 and L2 are never combined; instead, one weight regularization type out of L1, L2 and none is selected and we sample from the respective range afterwards. The parameters found to work best from the grid search were: agreement threshold of 1, 256 hidden units, 3 hidden layers, batch size 128, Adam with learning rate 1e-2, no batch norm, no dropout, L2-regularization with a coefficient of 1e-4, no L1-regularization. In practice, we often found it helpful to rescale the gradients after masking to compensate for the decreasing overall magnitude. We add the option for gradient rescaling as an additional hyperparameter, as we found it to help in several experiments. It rescales gradient components layer-wise after masking, by multiplying the remaining gradient components by $c$, where $c$ is the ratio of the number of components in that layer over the number of non-masked components in that layer (i.e. the sum of the binary elements in the mask).[14]. We speculate that for very large layers, a less extreme normalization scheme or the additional use of gradient clipping might be appropriate.

### B.3.2 DOMAIN ADVERSARIAL NEURAL NETWORKS

The experiments using DANN follow a similar pattern. The model consists of an embedding network, a classification network, and a "domain discrimination" network. All three modules are two-layer multi-layer perceptrons (MLP). The number of hidden units of all MLPs are sampled from the range specified in Table 1, and we trained 100 models. Both label classifier and domain discriminator are applied to the output of the embedding network. The label classifier is trained to minimize the cross-entropy-loss between the predicted and the true label. Similarly, the domain discriminator is trained to minimize the loss between predicted and true domain-label. The embedding network is trained to minimize the regular task classification loss and at the same time to maximize the the domain-loss achieved by the domain discriminator.

### B.3.3 INVARIANT RISK MINIMIZATION

For the experiments using IRM we used the authors' PyTorch implementation from `https://github.com/facebookresearch/InvariantRiskMinimization`. We perform a random hyperparameter search over with the ranges shown in Table 2

Table 2: Hyperparameter ranges for IRM.

| Hyperparameter | Ranges |
|---|---|
| No. hidden units | $\{256, 512\}$ |
| No. hidden layers | $\{3, 5\}$ |
| Batch-size | $\{64, 128, 256\}$ |
| Optimizer | $\{\text{Adam}_{\beta_1=0.9, \beta_2=0.999}, \text{SGD} + \text{momentum}_{0.9}\}$ |
| Batch-normalization | $\{\text{Yes, No}\}$ |
| Penalty weight | $\{10.0, 100.0, 1000.0\}$ |
| Number of annealing iterations | $\{0, 1, 2, 4, 8\}$ |
| Learning rate | $\{\text{1e-3, 1e-2, 1e-1, 1}\}$ |

### B.3.4 CURVES FOR ALL EXPERIMENTS

In Figure 14 we show the learning curves of training and test accuracy for the different methods.

### B.3.5 CORRELATION PLOTS

For the correlation plots in Figure 7 we used a randomly initialized MLP with the following configuration: 3 hidden layers, 256 hidden units. The dataset was using 16 environments and batches of size 1024. The lines in Figure 7 are linear least-squares regressions to the gradient data shown as scatter plots. We repeat the experiment 10 times with different network weight seeds, resulting in the 10 regression lines. Zero gradients are excluded from the regression computation, as most gradients are masked out by the product mask in both cases.

---

[14]Therefore, $c$ is 1 if the AND-mask has only 1s, and infinite if all components are masked out (which we then keep as 0.)

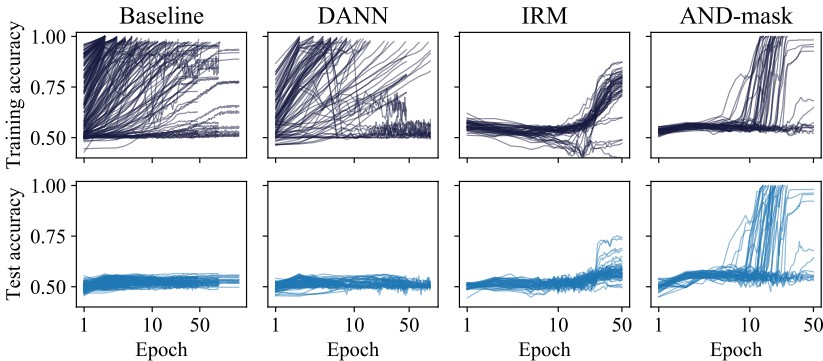

Figure 14: Learning curves for the evaluated methods. The top row shows the accuracy on the training set, the bottom row shows the accuracy on the test set.

## B.4 FURTHER VISUALIZATIONS AND EXPERIMENTS

In Figure 15 we show how many environments need to be present for the baseline without AND-mask to switch the decision boundary from the shortcuts to the mechanism. Under the same experimental condition as in the main paper, the baseline first succeeds at 1024 environments.

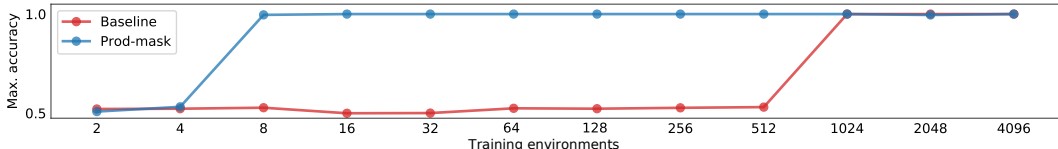

Figure 15: Relationship between number of training environments and test accuracy for the AND-mask method compared to the baseline. We show the best performance out of five runs using the settings that were used for the experiment in the main text.

## B.5 SECTION 3.2: CIFAR-10 MEMORIZATION AND LABEL NOISE EXPERIMENTS

**Memorization experiment** In Figure 16, we report the test performance (dashed lines) corresponding to the curves presented in the main paper for the CIFAR-10 memorization experiment. The test performance with standard labels decreases slower than the training performance as the threshold increases, and they eventually reach the same value. This is consistent with the hypothesis that by training on the consistent directions, the AND-mask selects the invariant patterns and prunes out the signals that are not invariant.

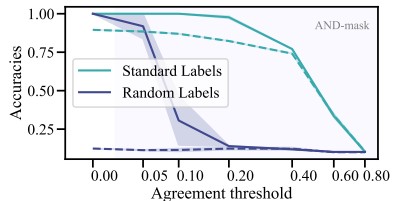

Figure 16: Dashed lines show test acc, solid lines show training acc.

**Network architecture and training details** Each trial trains the ResNet "`FastResNet`" from the PyTorch-Ignite example[15] for 80 epochs on the full CIFAR-10 training set. We use the Adam optimizer with a learning rate of $5e-4$, and a $0.1$ learning rate decay at epoch 40 and 60. We fix the batch size to 80. We set up 14 trials by evaluating each of the AND-mask-thresholds $\{0, 0.05, 0.1, 0.2, 0.4, 0.6, 0.8\}$ for two datasets: (a) unchanged CIFAR-10, (b) CIFAR-10 with the training labels replaced by random labels. Note that a threshold of 0 corresponds to not using the AND-mask. Each trial is run twice with separate random seeds.

**Label noise experiment** We trained the same ResNet as for the experiment above, once with and once without the AND-mask. We ran each experiment with three different starting learning rates

---

[15]`https://github.com/pytorch/ignite/blob/master/examples/contrib/cifar10/fastresnet.py`

$\{5e{-}4, 1e{-}3, 5e{-}3\}$ and a learning rate decay at epoch 60. The baseline worked best with a learning rate of $1e{-}3$, while the AND-mask with $5e{-}3$, likely to compensate for the masked out gradients. The AND-mask threshold that worked best was $0.2$, which is consistent with the results obtain in the experiment above.

### B.6 SECTION 3.3: BEHAVIORAL CLONING ON COINRUN

The target policy $\pi^*$ is obtained by training PPO (Schulman et al., 2017) for 400M time steps using the code[16] for the paper Cobbe et al. (2020). This policy is trained on the full distribution of levels in order to maximize its generality. We use $\pi^*$ to generate a behavioral cloning (BC) dataset, consisting of pairs $(s, \pi^*(a|s))$, where $s$ are the input-images ($64 \times 64$ RGB) and $\pi^*(a|s)$ is the discrete probability distribution over actions output by $\pi^*$.

The states are sampled randomly from trajectories generated by $\pi^*$. In order to test for generalization performance, the BC training dataset is restricted to 64 distinct levels. We generate 1000 examples per training level. The test set consists of 2000 examples, each from a different level which does not appear in the training set.

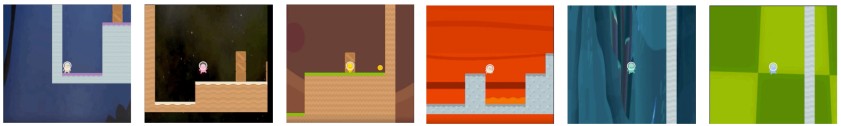

Figure 17: Screenshots of 6 levels of CoinRun (from OpenAI).

A ResNet-18 $\hat{\pi}_\theta$ is trained to minimize the loss $D_{\mathrm{KL}}(\pi^*||\hat{\pi}_\theta)$. We ran two automatic hyperparameter optimization studies using Tree-structured Parzen Estimation (TPE) (Bergstra et al., 2013) of 1024 trials each, with and without the AND-mask. The learning rate was decayed by a factor of 10 half-way at at $3/4$ of the training epochs.

The "temporal" version of the AND-mask used for this experiment is reported in Algorithm 1.

---

**Algorithm 1:** Temporal AND-mask Adam

1  $\mathbf{m} \leftarrow \beta_1 \cdot \mathbf{m} + (1 - \beta_1) \cdot \mathbf{g}$
2  $\mathbf{v} \leftarrow \beta_2 \cdot \mathbf{v} + (1 - \beta_2) \cdot (\mathbf{g} \circ \mathbf{g})$
3  $\mathbf{a} \leftarrow \beta_3 \cdot \mathbf{a} + (1 - \beta_3) \cdot \texttt{elemwise\_sign}(\mathbf{g})$
4  $\mathbf{b} \leftarrow \mathbb{1}[|\mathbf{a}| \geqslant \tau]$
5  $\theta \leftarrow \theta - \alpha(\mathbf{m} \circ \mathbf{b}) \oslash \sqrt{\mathbf{v} + \epsilon}$

---

In blue we highlight the additional lines compared to traditional Adam. The threshold $\tau$ and $\beta_3$ are hyperparameters that we included in the 1'024 trials of the search using Tree-structured Parsen Estimators. For the top 10 runs, hyperparameter values that were selected via the TPE search for the AND-mask are the following.

Table 3: Hyperparameters for the 5 best runs using the AND-mask, from the TPE search.

| Test KL div | lr | $\beta_1$ | $\beta_3$ | $\tau$ | weight decay |
|---|---|---|---|---|---|
| 1.652e-2 | 0.0078 | 0.21 | 0.79 | 0.36 | 0.057 |
| 1.656e-2 | 0.0072 | 0.26 | 0.86 | 0.40 | 0.041 |
| 1.662e-2 | 0.0080 | 0.23 | 0.84 | 0.41 | 0.045 |
| 1.665e-2 | 0.0068 | 0.33 | 0.72 | 0.47 | 0.077 |
| 1.672e-2 | 0.0063 | 0.67 | 0.65 | 0.47 | 0.080 |

We found that applying weight decay as a second independent update *after* the AND-mask routine improved performance. To keep the comparison fair, we added this as a switch in the hyperparameter search for the Adam baseline as well, and it improved performance there as well.

---

[16] https://github.com/openai/train-procgen

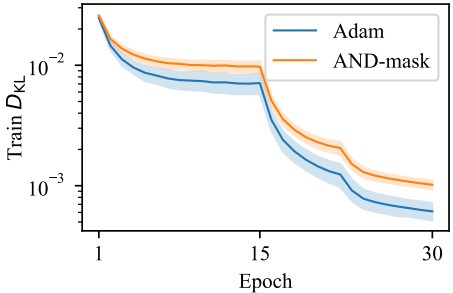 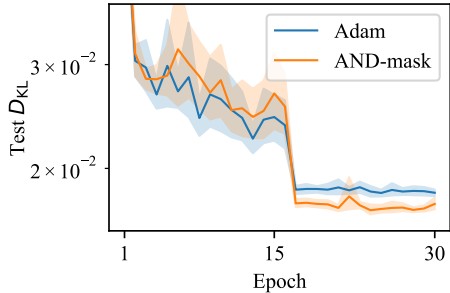

Figure 18: Learning curves for the behavioral cloning experiment on CoinRun. Training loss is shown on the left, test loss is shown on the right. We show the mean over the top-10 runs for each method. The shaded regions correspond to the 95% confidence interval of the mean based on bootstrapping.

## C  APPENDIX TO SECTION 4

### C.1  RELATED WORK IN CAUSAL INFERENCE

**Causal graphs and causal factorizations**    The formalization of causality through directed acyclic graphs (Pearl, 2009) is a key element informing our exposition. According to such formalization, a causal model gives rise to each observed distribution. It is thereby possible to exploit properties of the causal factorization of the joint probability distribution over the observed variables. Clearly, there are many ways to factorize a joint distribution into conditionals; a distinguishing feature of the causal factorization is that many of the conditionals, which we can think of as physical mechanisms underlying the statistical dependencies represented, are expected to remain *invariant* under interventions or changing external conditions. This postulate has appeared in various forms in the literature (Haavelmo, 1943; Simon, 1953; Hurwicz, 1962; Pearl, 2009; Schölkopf et al., 2012).[17]

**Causal models and robust regression**    Based on this insight, it was proposed that regression based on causal features should presents desirable invariance and robustness properties (Mooij et al., 2009; Schölkopf et al., 2012; Peters et al., 2016; Rojas-Carulla et al., 2018; Heinze-Deml et al., 2018; von Kügelgen et al., 2019; Parascandolo et al., 2018). In this view, the mechanisms can be considered as features of the patterns such that they support stable conditional probabilities. Thus learning the mechanisms may help achieve a stable performance across a number of conditions. Other works connecting causality and learning through invariances are (Subbaswamy et al., 2019; Heinze-Deml and Meinshausen, 2017), and perhaps – most related to our work – (Arjovsky et al., 2019): we presented a comparison with this method in the following section.

**Causal regularization**    Recently (Janzing, 2019) showed that biasing learning towards models of lower complexity might in some cases be beneficial for a notion of generalization from observational to interventional regimes. Our proposed solution is however different, in that we only indirectly deal with penalizing model complexity, and rather focus on our proposed notion of consistency.

### C.2  LEARNING INVARIANCES IN THE DATA

Here we are going to compare ILC to other approaches for learning invariances in the data with neural networks, and in particular to Invariant Risk Minimization (IRM) Arjovsky et al. (2019). The authors of IRM analyze a set up where minimizing training error might lead to models which absorb all the correlations found within the training data, thus failing to recover the relevant causal explanation. They consider a multi-environment setting and focus on the objective of extracting data representations that lead to invariant prediction across environments.

While the high level objective is close to the one we focused on, the differences become clear when considering the definition of *invariant predictors* presented in Arjovsky et al. (2019):

---

[17]This would be different for a non-causal factorization of the joint distribution, see Schölkopf (2019)

*Definition* 1. A data representation $\Phi : \mathcal{X} \to \mathcal{H}$ elicits an invariant predictor $w \circ \Phi$ across environments $\mathcal{E}$ if there is a classifier $w : \mathcal{H} \to \mathcal{Y}$ simultaneously optimal for all environments, i.e., $w \in \arg\min_{\bar{w}:\mathcal{H}\to y} R^e(\bar{w} \circ \Phi) \; \forall e \in \mathcal{E}$.

In particular, the objective minimized by IRM is:

$$\min_{\Phi:\mathcal{X}\to\mathcal{Y}} \sum_{e\in\mathcal{E}_{\mathrm{tr}}} R^e(\Phi) + \lambda \cdot \left\| \nabla_{w|w=1.0} R^e(w \cdot \Phi) \right\|^2 \tag{5}$$

where $\Phi$ are the logits predicted by the neural network and $w$ is a dummy scaling variable (see Arjovsky et al. (2019)). The relevant part is the penalty term $\lambda \cdot \left\| \nabla_{w|w=1.0} R^e(w \cdot \Phi) \right\|^2$: One way to interpret it, is that the penalty is large on every environment where the distribution outputted by $\Phi$ could be made 'closer' to the distribution of the labels by either sharpening ($w > 1$) or softening it (i.e., closer to uniform $w < 1$).

Let us consider the example from IRM, where the authors describe two datasets of images that each contain either a cow or a camel: In one of the datasets, there is grass on 80% of the images with cows, while in the other dataset there is grass on 90% of them. IRM then makes the point that we can learn to ignore grass as a feature, because its correlation with the label cow is inconsistent (80% vs 90%). The setting we consider in this paper is slightly different: take our example from the CIFAR-10 experiments. Under our concept of invariance, we expect that (depending on the data generating process) even a single dataset where we treat every image as coming from its own 'environment' should be sufficient to discover invariances. Drawing a connection to the setting from IRM, we would argue that the second dataset should not be necessary to learn that 'grass' is not 'cow'. If one treats every example as coming from its own environment, there is already sufficient information in the first dataset to realize that cows are not grass: Grass is predictive of cows only in 80% of the data, so grass cannot be 'cow'. The actual cow on the other hand, should be present in 100% of the images, and as such it is the invariance we are looking for. Note that this is of course a much more strict definition of invariance: If our dataset contains images labeled as 'cows' but that have no cows within them, we might start to discard the features of cows as well.

