# OpenReview forum: "Learning explanations that are hard to vary"
_ICLR.cc/2021/Conference — ICLR 2021 Poster_

### Official Review · AnonReviewer3 · 2020-10-27
**Ingenious approach to learn invariant solutions across environments but connections to causality are too vague**

**Rating:** 5
**Confidence:** 4

**Review:**

The authors attempt to recover invariant solutions, defined as those with a similar loss neighbourhood across environments. This concept is interesting, especially since it can be incorporated with a clever understanding of loss gradients and hessians which underlies the generality of the proposed approach.

The presentation is puzzling to me. The objective for all intents and purposes is to recover the causal solution from data in multiple environments, but this is never explicitly mentioned even though causality is said to be a "key element" in the Appendix. In this area, also precise conditions are needed on data distributions from different environments which seem to play no role in this paper. A more thorough description of what can be expected from invariant solutions and when we are expected to consistently recover them is needed in my opinion. The introduction is therefore also vague, what do we mean by generalizing out-of-sample? Is the objective to ensure good performance with arbitrary shifts in distribution while maintaining a causal graph? Is there an assumption that the invariant mechanism is unique among available environments?

Regarding the ILC, hessians and gradients I do see they are related but it is not lear to me how changing hessians would lead to optimizing for the ILC. The ILC is defined with respect to all local minima, I am not sure how such an objective would be attainable. More details here would be helpful.

In general, I found all experiments highly constructed and nothing like what you'd expect to see in reality. If the objective is improved performance out-of-sample, why not evaluate the method on domain generalization tasks (office, caltech datasets, etc.)? Invariant solutions may also be defined as minimizing a worst-case optimization problem across environments, yet the robust optimization literature is not mentioned.

Minor comments:
- I may be misunderstanding, but as described in the Appendix, does the synthetic experiment have shortcuts for both labels sampled from the same distribution?
- With unobserved confounding, it is known that the causal solution does not have zero loss gradient (for example in a linear model, least squares returns biased parameter estimates), what can we expect in that case?


In summary, I do believe the proposed approach to be interesting but the context in which it is expected to work, as well as its relationships with causality, robust optimization and domain generalization are not well described; this prevents me from raising my score at this moment.

---

> ### Author Response · Authors · 2020-11-18
> **R3 response**
>
> Thank you for the constructive feedback! We answer your concerns in the following:
>
> **Connection to causality:** The algorithm and derivation we propose in the paper do not rely on assumptions from causality and as such we provide no guarantee that causal variables will always be recovered. Gradients (and Hessians) are strongly influenced by the inductive biases introduced by the choice of neural networks, and the method relies on these to define what is invariant and what is not.
> Moreover, given that we consider neural networks and a nonlinear setting, a theoretical characterization of identifiability (elsewhere proved based on strong assumptions such as, e.g., linearity of the model and Gaussianity of the errors, see Peters et al., 2015) is highly nontrivial for our setting.
> Nevertheless, our work lends itself to be interpreted in a causal framework:
> From the point of view of causal modeling, $f$ can be thought of as a conditional distribution of the targets given causal features of the inputs; invariance of such conditionals is expected if they represent causal mechanisms (e.g., Hoover, 1990), that is --- stable properties of the physical world. Generalizing out-of-sample means therefore that the predictor should perform equally well on data coming from different settings, as long as they share the causal mechanisms.
> Our current work can be placed within the recent research direction of learning invariances and ood generalization -- such as Invariant Risk Minimization (Arjovsky et al., 2019), Risk Extrapolation (Krueger et al., 2020), IRM games (Ahuja et al., 2020) and generalization/memorization in neural networks (e.g. Understanding deep learning requires rethinking generalization, Zhang et al, 2017). A more formal study of our algorithm within the framework of causality -- with clear assumptions on the SCMs generating the data -- is an interesting direction for future work.
>
> **With unobserved confounding, it is known that the causal solution does not have zero loss gradient (for example in a linear model, least squares returns biased parameter estimates), what can we expect in that case?:**
> Thanks for this interesting question!
> While an investigation of our method in a setting with linear models and unobserved confounding is possible and valuable, we believe that an in-depth theoretical and empirical analysis deserves its own consideration and goes beyond the scope of this paper. Nevertheless, we ran a simple and very preliminary experiment based on the setting you suggest, i.e. a linear regression problem with confounding with three different causal graphs. We attach the code as a jupyter notebook in the folder **"causal_graph_example_for_Reviewer3"**.
> The target invariant solution (i.e. the one using causal variables only) should be w = [1, 0] in all three graphs (but in the last graph the second variable is only a dummy variable). Training with ERM (classic arithmetic mean of the gradients) converges to biased solutions in all three cases ((A): [1, 0.33], (B): [0.66, 0.33], (C): [1.33, 0.]), while training with the and-mask as presented in the paper converges to [1, 0] on all three problems.
> We find these very preliminary results encouraging and an interesting direction for future investigations.
>
> **The ILC is defined with respect to all local minima, I am not sure how such an objective would be attainable. More details here would be helpful:**
> The derivation of the algorithm we propose is the result of a series of approximations to go from the original definition of inconsistency and ILC to the and-mask. Summarized, these are: ILC -> inconsistency of the minima -> hessians -> diagonal hessians -> geometric mean of gradients -> and-mask. We hope that future work will derive new algorithms using different approximations, e.g. optimizing the ILC objective using meta-learning.
>
> **If the objective is improved performance out-of-sample, why not evaluate the method on domain generalization tasks (office, caltech datasets, etc.)?**
> We believe that strong performance on visual tasks is conditional on the presence of features optimized on large amounts of data or with clever inductive biases specific to images (such as data augmentation techniques and self supervised learning). With a poor inductive bias several spurious but invariant explanations might emerge regardless of the algorithm. Very recent work by Gulrajani and Lopez-Paz (2020) "In Search of Lost Domain Generalization" showed that in these tasks ERM is still as good or even better than most methods presented in the literature on OOD-generalization and domain adaptation. For this reason, we focused on tasks where we could study a purely algorithmic improvement. For example, the choice of a synthetic but challenging dataset allowed us to intervene on the data-generating process, and we could perform interesting experiments such as the one with the XOR and AND masks.
>
> (continues)

---

> > ### Author Response · Authors · 2020-11-18
> > **(part 2)**
> >
> > **Does the synthetic experiment have shortcuts for both labels sampled from the same distribution?:**
> > For each environment we sample one shortcut, as a vector $\mathbf{x}_s$ from a multivariate Gaussian distribution with diagonal covariance. We then use that vector $\mathbf{x}_s$ as the shortcut for class 1, and its opposite $-\mathbf{x}_s$ for class -1. We put all details in Appendix B.2.
> >
> > **Connections to domain generalization, robust optimization:**
> > As discussed in the related work section, while our approach can indeed be used in the context of domain generalization, it can be applied to settings that go beyond those allowed by most methods from that literature (like the CIFAR example and DANN with 50'000 classes).
> > Regarding robust optimization (RO): thank you for pointing out this interesting connection. We are not aware of approaches from RO that would help to solve the task we study in our paper. To our understanding, based on Ben-Tal et al., RO is usually restricted to convex problems and violations of constraints. Instead, tasks explored in our paper are nonconvex and, most importantly, unconstrained. If you are aware of any literature on RO which might be useful for our setting, we would be very glad to incorporate a remark.
> >
> > We are glad you found overall the approach interesting, and that it provides a clever understanding of its connection to gradients and Hessians in different environments. We are happy to address any remaining concerns you might have.

---

### Official Review · AnonReviewer4 · 2020-10-30
**This paper studies learning invariances to improve generalization of models. The paper is novel and solid, thus I suggest acceptance of the paper.**

**Rating:** 7
**Confidence:** 3

**Review:**

This paper investigates one possible pitfall of current gradient descent method that averaging gradients over different examples failed to capture the invariance between different examples, through it can learn quickly by memorizing data. The authors claim that this is one important reason that machine learning with GD can't generalize well to out of domain dataset. To solve this issue, this paper focus on developing a method to learning the invariant explanations. The authors firstly formalize the a notion of consistency of the loss surface and also propose a practical algorithm, AND-mask. The authors also carried experiments and analysis on synthetic dataset to validate the effective the method. Overall, the work is important and the paper is well presented. I vote for accepting.


##########################################################################
Pros
	1. The paper studied the important problem of model generalization. The paper is well motivated and the knowledge presented can benefit the community.
	2. The paper defines the invariance as the consistency for minima of the loss surface. The author presented case study examples as well as math formulations. The presentation is clear and easy to understand.
	3. The authors validate the value of capturing invariance through the implementation of a practical AND-mask, and run experiment on a set of tasks. The baselines are reasonably chosen. The proposed method greatly outperforms other baselines.
	4. The synthetic tasks unveil the difference between ILC and gradient averaging. It is very helpful for the readers.

##########################################################################
Cons
	1. The work claim that they run experiments on a set of real-world tasks. The authors introduce artificial label noisiness in CIFAR-10 by randomly shuffling the labels. However, recently work (e.g https://arxiv.org/pdf/2001.10528.pdf) found that noisiness exists in many popular datasets. It would be interesting to show that ILC can also handle such naturally introduced noises.
2. What is the connection between invariance and model casual explanation?

---

> ### Author Response · Authors · 2020-11-18
> **R4 response**
>
> **(1)** Thanks for pointing out the recent work from arxiv.org/pdf/2001.10528 and their interesting findings regarding noisiness in popular datasets. We believe the direction you suggest is a promising avenue for future work, which we hope will be explored in the near future. For the scope of this paper we picked the memorization and label noise tasks for CIFAR-10, since they have been extensively used in recent works and are more straight-forward to investigate.
>
> **(2)** Regarding the connection between invariance and causal explanations, the perspective we take in the paper is the following: invariances are likely to be present in the data due to causal models underlying the data generating process. Our work can be placed within the recent research direction of learning invariances and ood generalization -- such as Invariant Risk Minimization (Arjovsky et al., 2019), Risk Extrapolation (Krueger et al., 2020), IRM games (Ahuja et al., 2020) and generalization/memorization in neural networks (e.g. Understanding deep learning requires rethinking generalization, Zhang et al, 2017). As you can see the derivation of the algorithm itself does not rely on any formal assumptions from causality.
>
> Thank you for your comments, we are very glad you found the paper to be novel and solid, that it studies an important problem, and that it is well motivated, easy to understand, and clearly presented. We are also glad you found the tasks we chose helpful to the readers. Do you have any concerns left that we might answer?

---

### Official Review · AnonReviewer2 · 2020-11-05
**Interesting**

**Rating:** 2
**Confidence:** 3

**Review:**

this work posits that invariant mechanisms exist in a dataset. a machine learning algorithm that is trained using gradient descent usually averages gradients across examples. the thesis is that by averaging gradients, information is lost. the method posits that in a gradient descent algorithm, instead of an arithmetic average, a geometric (or karcher) mean can be used to preserve information about invariant mechanisms - while ignoring confounders. there are difficulties in a straightforward application of the geometric mean, so a simple heuristic algorithm is developed, involving masking gradients depending on whether the sign of the gradient agrees across a batch of examples (or, whether some agreement threshold is reached). this algorithm is tested on a synthetic dataset, a semi-synthetic task on CIFAR-10, and coinbase, an RL algorithm.

recommend major revisions and baselines. it is interesting ideas and it would do the ideas justice to put a lot of effort into a rewrite so the ideas are properly understood by the ICLR community. clarity needs to be significantly improved in the introduction and conceptual framing (mainly in intro/methods sections)—the rest of the paper is largely well-written, the experiments are well-documented in the appendix, relationship to invariant risk minimization and causality is documented in appendix.

baselines: sum vs geometric mean is a well-known effect in products of experts vs sums of experts work. i think this should be the right baseline to compare to. train separate classifiers and use them as a product of experts, and you should get the same performance as in this paper. definitely need to discuss this related work and how learning consistency /geometric mean measures differ and warrant comparison.

the writing is more difficult to read than it ought to be. more work is needed to re-use existing concepts. for example, consistency is overloaded here (it might confuse some readers from a statistics background). the paper's title has the concepts of varying and explanations (variance and explanations do not appear in the method). small things like this abound. trying to find specific examples where i have a strong opinion to help:

* instead of consistency, it may be worth spending some time to think of another word that is less confusing. congruence? concordance? 'learning invariant mechanisms with geometric mean gradient descent' is an example of a title that would be easier to understand: (1) mechanisms relates to prior work such as the pearl work that is cited (indeed, in the appendix the authors state 'causality... is a key element informing our exposition'). a causal mechanism is a well-known concept to the ICLR community. (2) concordance is still overloaded (e.g. this is a major concept in biostatistics) but not to the degree that consistency is. similarly, if you choose congruence, this could have the nice geometric interpretation in the hessian example that is helpful for understanding the work in figure 3.

* the david deutsch quote distracts and confuses, remove it (i also know some researchers that might be immediately skeptical of a david deutsch citation in an ICLR paper). see above for ideas on title changes.

* the first example is helpful to understand the goals of the paper. however, figure 1 --  the symbol with the plus sign and arrow is confusing, all the axes tick labels need to be much larger

* the 'an example' chess example in the gray box on page 2 is unnecessary and distracting. instead, the discussion of the relationship to arjovsky et al could be inserted (with grass and cows), which is much more interesting and useful to a reader.

* equation 1 engenders a double negative throughout the conceptual framing of the paper. i highly suggest considering renaming this to a 'consistency' score to reflect the name of the method; this will make it easier for readers to understand.

* 'patchwork' may be another jargon word, try to find another one that is easier to understand. on page 3: 'low consistency of a classic patchwork solution' is loaded with jargon. first, a patchwork solution is not defined here or in previous work. second, a 'classic patchwork' solution is more confusing. removing adjectives such as 'classic' or even 'patchwork' throughout the paper may help readability and clarity.

* i found the synthetic memorization dataset very difficult to understand. never start a sentence with math like p(y | x_{d_M}), it looked like it was connected to the previous with a \cdot. ideally write out the functional form of the mechanism. the axes in figure 5 are not labeled, so i have a hard time understanding what we are looking at: are the x and y axes d_S or d_M? how is environment A vs environment B defined?

* re: wording -- ideally delete shortcuts. do not introduce more unnecessary lingo, because it leads to sentences like 'the shortcuts are not shared across environments, but provide a simple way to classify the data, even when pooling all the environments together'. this is asking a reader to do a lot of work: (1) what is pooling? this has a standard usage. (2) what are environments? this is used in RL (3) what are shortcuts, and how do they relate to environments? trying to rewrite this, i would want something with fewer novel concepts, that focuses on mechanisms that are invariant across datapoints or groups of datapoints (groups of data arising from causal graphs, e.g. in causal inference terminology).

* extremely minor, a few typos and inconsistent capitalization (section 3.1 is not capitalized, figure titles not capitalized, paragraph headings, figure legends not capitalized , good to pick one and stick to it)

---

> ### Author Response · Authors · 2020-11-18
> **R2 response**
>
> We are glad to see you found the method interesting, the experiments well documented, the connection to IRM and causality properly addressed, and the paper largely well written, except for some presentation issues in the introduction and method.
> However, we are having a hard time reconciling the review with the initial score of 2, a strong rejection score, especially since all reviews highlight contributions that are valuable to the ICLR community.
>
> Based on your suggestions, we changed the look of several figures and rephrased sentences starting with mathematical symbols.
>
> We decided to keep terms such as "environments", "shortcuts", "pooling data together", as they appear in several recent works on learning invariances and o.o.d. generalization (e.g. Invariant Risk Minimization (Arjovsky et al., 2019), "Out-of-Distribution Generalization via Risk Extrapolation" (Krueger et al., 2020), "Shortcut Learning in Deep Neural Networks" (Geirhos et al, 2020), "What shapes feature representations? Exploring datasets, architectures, and training" (Hermann & Lampinen, 2020), etc.).
>
> Thank you for pointing out Product of Experts. It is unclear to us that PoE in the form you suggest would get the same performance obtained by the methods proposed in this paper, since taking the product of the outputs of overfitted classifiers will not recover the invariant solution. Consider that by training on any one (and one only) of the environments in the synthetic dataset, standard classifiers learn the simple decision boundary instead of the invariant mechanism (which cannot even be identified as invariant if there are no other environments).
> Moreover, there would be practical problems as well: for example in the CIFAR experiments, based on your proposal of training one classifier per environment (i.e. here example), one would have to train 50'000 classifiers (which is a similar problem shared by DANN).

---

### Official Review · AnonReviewer1 · 2020-11-10
**Evaluations of Paper "Learning explanations that are hard to vary"**

**Rating:** 9
**Confidence:** 4

**Review:**

The authors introduce and formalize the concept of Invariant Learning Consistency (ICL), which is motivated by the idea that "good explanations are hard to vary" in the context of deep learning. Instead of using the arithmetic mean to pool gradients (logical OR), the authors propose to use the element-wise geometric mean of gradients with a logical AND masking. Experimental results on both synthetic and real-world data sets are reported under the setup of supervised learning and reinforcement learning.

This paper is well written and clearly presented. The exploration of using geometric mean with a logical AND masking to pool gradients in deep learning is very interesting and novel. The proposed method learns invariances in gradient-based optimizations and can help with the memorization issue. This work has made good efforts towards a better understanding of learning, memorization and generalization of OOD. Experimental results on both synthetic and real-world data sets have demonstrated the effectiveness of the proposed AND-mask, comparing with commonly used regularizers and several baselines.

As the authors point out, the AND-mask is one of multiple possible ways to improve consistency, and it is unlikely to be a practical algorithm for all applications. It would be great if the authors can provide further insights on what kind of data distributions (or applications) could potentially benefit from the proposed method versus using the arithmetic mean. In addition, it would be interesting to explore hybrid methods that combine the advantages of pooling gradients using arithmetic mean and geometric mean (i.e., AND-mask).

---

> ### Author Response · Authors · 2020-11-18
> **R1 response**
>
> Thank you for the positive review. We are glad you found the paper very interesting, well written, novel, and properly evaluated.
>
> **As the authors point out, the AND-mask is one of multiple possible ways to improve consistency, and it is unlikely to be a practical algorithm for all applications. It would be great if the authors can provide further insights on what kind of data distributions (or applications) could potentially benefit from the proposed method versus using the arithmetic mean:**
> This is an interesting question. We believe that taking the arithmetic mean is suboptimal in situations where it is important to avoid finding spurious solutions. A potential application for our framework might be in vision for autonomous driving systems, e.g. where the change in colours of nearby cars should not affect the driving.
> At the same time, we want to emphasize that large "OR" representations might not necessarily be worse in every setting. There might be settings where instead of learning only the invariant mechanisms, it could be best to have a large bag of (even partially) relevant features. This leads to your next point:
>
> **In addition, it would be interesting to explore hybrid methods that combine the advantages of pooling gradients using arithmetic mean and geometric mean (i.e., AND-mask):**
> We agree, in practice what turns out to be most useful might be to learn both the "OR" and the "AND" representations, and rely on either of the two (or both) depending on the setting. While our threshold parameter attempts to build a bridge between the arithmetic mean and geometric mean gradients, the space of possibilities is ample for future work in this direction (early layers with OR and later layers with AND, pre-training with OR and fine-tuning with AND, etc.).

---

### Author Response · Authors · 2020-11-18
**To all reviewers**

We thank all reviewers for their constructive feedback.

In an effort to improve reproducibility and to allow potential future work to build on ours, we have released an implementation of the and mask and geometric mean as presented in this paper, which is now attached as supplementary material.

---

### Decision · Program_Chairs · 2021-01-07
**Final Decision**

**Decision:**

Accept (Poster)

**Comment:**

The paper shows that hat if the goal is to find invariant mechanisms in the data, these can be identified by finding explanations (e.g. model parameters) that are hard to vary across examples. To find those "explanations" it then proposes to combine gradients across examples in a "logical AND" fashion, i.e., pooling gradients sing a geometric mean with a logical AND masking. All reviewers agree that the direction is very interesting. While indeed mentioning sum and products of experts might be good, the overall idea is still very much interesting, also to the ICRL community, since it paves the way to apply this to larger set of machine learning methods, as actually shown in the experimental evaluation. Still, the authors should make the link to causality more obvious from the very beginning. This should also involve clarifying that "explanations" here do not refer to "explanations" as used in Explainable AI. Overall, this is an interesting and simple (in a positive sense) contribution to the question of getting at least "more" causal models.